# Cross-validation for Geospatial Data: Estimating Generalization Performance in Geostatistical Problems

**Jing Wang**  *wangji9@oregonstate.edu*
*School of Electrical Engineering and Computer Science*
*Oregon State University*
*Corvallis, OR 97331-5501, USA*

**Laurel M. Hopkins**  *hopkilau@oregonstate.edu*
*School of Electrical Engineering and Computer Science*
*Oregon State University*
*Corvallis, OR 97331-5501, USA*

**Tyler A. Hallman**  *t.hallman@bangor.ac.uk*
*School of Natural Sciences*
*Bangor University*
*Bangor LL57 2DG, UK*

**W. Douglas Robinson**  *douglas.robinson@oregonstate.edu*
*Department of Fisheries, Wildlife, and Conservation Sciences*
*Oregon State University*
*Corvallis, OR 97331-5501, USA*

**Rebecca A. Hutchinson**  *rah@oregonstate.edu*
*School of Electrical Engineering and Computer Science*
*Department of Fisheries, Wildlife, and Conservation Sciences*
*Oregon State University*
*Corvallis, OR 97331-5501, USA*

**Reviewed on OpenReview:** *https://openreview.net/forum?id=VgJhYu7FmQ*

## Abstract

Geostatistical learning problems are frequently characterized by spatial autocorrelation in the input features and/or the potential for covariate shift at test time. These realities violate the classical assumption of independent, identically distributed data, upon which most cross-validation algorithms rely in order to estimate the generalization performance of a model. In this paper, we present a theoretical criterion for unbiased cross-validation estimators in the geospatial setting. We also introduce a new cross-validation algorithm to evaluate models, inspired by the challenges of geospatial problems. We apply a framework for categorizing problems into different types of geospatial scenarios to help practitioners select an appropriate cross-validation strategy. Our empirical analyses compare cross-validation algorithms on both simulated and several real datasets to develop recommendations for a variety of geospatial settings. This paper aims to draw attention to some challenges that arise in model evaluation for geospatial problems and to provide guidance for users.

## 1 Introduction

A variety of geospatial problems benefit from machine learning (ML) analyses. Of note, many problems with societal and environmental importance have geospatial properties. For example, machine learning models support crop production in agricultural applications (Dadsetan et al., 2021), predictions of poverty

(Jean et al., 2016), analysis of forest fire risk (Yang et al., 2021), and creation of land cover maps (Bjorck et al., 2021) across space. As a specific motivating example, consider the problem of species distribution modeling (SDM). Species distribution models capture relationships between species and their habitats based on observations at a set of locations and the associated environmental variables at those locations (Elith & Leathwick, 2009). SDMs are critical tools for ecologists and natural resource managers seeking to model the current, potential, and future distributions of threatened species (e.g., Wilson et al., 2021; Hallman et al., 2021). Machine learning approaches like random forests, boosted regression trees, and maximum entropy models have been shown to perform well on this task (Valavi et al., 2021); these methods are flexible and powerful but not explicitly spatial models. They are typically fit with only environmental features and not spatial coordinates as inputs. Some key common characteristics of the geospatial problems we focus on in this paper are: 1) they fit models to and predict the response variable at geolocated data points; 2) despite the spatial nature of the data, practitioners may choose to apply models that do not explicitly account for space, instead favoring easy-to-use, flexible, non-spatial techniques; 3) the intended test region and its environmental features may be known or estimable.

Just as in non-spatial domains, geospatial applications rely on estimates of generalization performance for model evaluation and selection. Classical cross-validation (CV) techniques estimate generalization performance by dividing the training set into folds; by default, they assign data points to folds uniformly at random. Each fold takes a turn acting as a validation fold on which performance is measured after training a model on the other folds. These measurements estimate how well a model will perform when applied to a new, unseen dataset (i.e., the test dataset). These performance estimates may have stand-alone value; e.g., a natural resource manager may only wish to proceed with some conservation action based on a sufficiently high-quality SDM. Performance estimates may also serve model selection goals; e.g., an ecologist may want to select the best of a set of modeling approaches for further study.

Geospatial problems pose unique challenges to standard CV approaches. In particular, theoretical results showing the unbiasedness of the standard CV estimators assume independent, identically distributed (iid) data (Arlot & Celisse, 2010; Hoffimann et al., 2021). However, *spatial autocorrelation* among features (i.e., feature values at geographically proximal points are more similar than feature values at geographically distant points) in geospatial contexts induces a correlation structure among CV folds, so data points in the training and validation folds may violate the assumption of independence. Furthermore, *covariate shift* is common in geospatial problems (i.e., the distribution of features may change across space), so data points in the training and validation folds may violate the assumption of being identically distributed. In a given problem, either or both of these challenges may be present to varying degrees. For example, in one SDM application in Sec. 7.3.1 below, we applied CV to models built with a training set comprised of bird observations at two national parks to estimate performance on a third, geographically distinct, national park. Prediction points in the test park are not spatially autocorrelated with those in the training parks, though the points within the training parks are spatially autocorrelated with each other. In addition, the feature distributions at the test park differ from that of the training parks. Therefore, this problem does not meet the iid assumption required to guarantee unbiased performance estimates from standard CV techniques. Some extensions to these techniques have been developed to address spatial autocorrelation and covariate shift (reviewed below), but an approach to address both simultaneously is still lacking (Hoffimann et al., 2021).

In this paper, we report on an investigation into how spatial autocorrelation and covariate shift combine in different geospatial learning settings to affect generalization performance estimates from a variety of CV strategies. Our specific contributions are:

- We prove a theoretical criterion for unbiasedness of CV estimators in settings with spatial autocorrelation and/or covariate shift.

- We explore criteria for sorting problems into geospatial scenarios to aid in selection of an appropriate CV approach for a given analysis.

- We propose a new CV algorithm to address spatial autocorrelation and covariate shift simultaneously.

- We provide simulated data experiments measuring the bias of several CV estimators, SDM examples demonstrating four geospatial scenarios, and further empirical analyses of our proposed algorithm.

The rest of the paper is organized as follows. Section 2 gives background on cross-validation and reviews related work. Section 3 defines the geospatial problem setting formally. Section 4 proves a criterion for unbiased CV estimators in geospatial settings. Section 5 explores a framework for categorizing geospatial problems into four discrete scenarios, based on whether they display spatial autocorrelation and/or covariate shift. Section 6 introduces a new cross-validation procedure aimed at geospatial settings in which both spatial autocorrelation and covariate shift are at play. Section 7 reports on three sets of experiments to evaluate a set of CV procedures. Section 8 discusses the insights and recommendations we can glean from this work so far. Section 9 concludes with a summary and some ideas for future research.

## 2   Background & Related Work

First, let us be clear about how we quantify generalization performance. Test error and risk are two common statistics in model evaluation (Hastie et al., 2001). Test error ($Err_T$) is the expected loss over test samples, given a training set $T$ of $n$ examples. Risk ($R$) is the expected test error over training sets from the same population. The critical difference between these is whether an expectation is taken over training sets (risk), as opposed to conditioning on a single training set (test error). Hereafter, $tr$ and $te$ subscripts denote training and test, respectively, and $i$ and $j$ subscripts index training and test samples, respectively. We use $X$ and $\mathbf{x}$ for features and $\mathbf{y}$ and $y$ for response variables. Then for a general loss function $\mathcal{L}$, test error and risk are defined as

$$Err_T^{(n)} \equiv \mathbb{E}_{T_j}[\mathcal{L}(y_j, \hat{y}(\mathbf{x_j}; T))|T], \tag{1}$$

$$R^{(n)} \equiv \mathbb{E}_{T,T_j}[\mathcal{L}(y_j, \hat{y}(\mathbf{x_j}; T))], \tag{2}$$

where $T_j = \{\mathbf{x_j}, y_j\}$ is a test sample randomly drawn from the joint distribution $P_{X_{te}, \mathbf{y_{te}}}$, $T = \{\mathbf{x_i}, y_i\}_{i=1}^n$ is a training set randomly drawn from the joint distribution $P_{X_{tr}, \mathbf{y_{tr}}}$, and $\hat{y}(\mathbf{x_j}; T)$ is the prediction of $y_j$ given $\mathbf{x_j}$ and $T$.

Cross-validation assesses a model's predictive capabilities. Its target is sometimes misunderstood to be an estimate of test error, but under the iid assumption, it actually estimates the risk (Bates et al., 2023). Hastie et al. (2001) conclude that the estimation of test error for a particular training set is not easy in general, but that cross-validation may provide reasonable estimates of risk. Both risk and test error may be of interest to practitioners, in terms of expected performance on testing sets drawn from some population. Risk estimates speak to a model's expected performance when fit to training sets drawn from some population, whereas test error estimates speak to a model's expected performance when fit to a particular training set. Risk is amenable to theoretical analysis and simulation experiments, and this is the quantity of focus for those sections below. However, when only a single training set is available, as in our empirical analyses, only test error is estimable. The discrepancies between the risk and test error perspectives are a challenging aspect of research on cross-validation.

Standard techniques for cross-validation put data points into folds uniformly at random. $K$-fold cross-validation (KFCV) randomly divides a training set $T$ into $K$ non-overlapping folds and iteratively holds out one fold at a time, training a model on the remainder (training folds $T_{-k}$) and measuring error on the held-out fold (validation fold $T_k$). The average of these model errors across folds is the estimate of risk. Leave-one-out cross-validation (LOOCV) is $n$-fold CV, in which the validation fold is a single data point, $T_i$, and the training fold contains the remaining data, $T_{-i}$. The corresponding risk estimates for these approaches are

$$\hat{R}_{KFCV}^{(n)} \equiv \frac{1}{n} \sum_{k=1}^{K} \sum_{i \in T_k} \mathcal{L}(y_i, \hat{y_i}(\mathbf{x_i}; T_{-k})),$$

$$\hat{R}_{LOOCV}^{(n)} \equiv \frac{1}{n} \sum_{i=1}^{n} \mathcal{L}(y_i, \hat{y_i}(\mathbf{x_i}; T_{-i})).$$

The bias-variance trade-off in KFCV is controlled by the user-defined parameter $K$ (Arlot & Celisse, 2010). In general, bias decreases while variance increases as $K$ increases. To see this, note that LOOCV builds $n$ models, each of which is trained on $n-1$ samples, while KFCV builds $K$ models, each of which is trained on $(K-1)n/K$ samples. Since the size of the training folds under LOOCV is closer to the full dataset size

than in KFCV, LOOCV incurs less bias. However, LOOCV is more computationally expensive than KFCV when $n \gg K$. For iid data, these standard CV estimators are unbiased; their random resampling mechanism mimics the way that a new sample would be collected from the population.

In addition to these standard approaches, a few spatial cross-validation procedures have been developed to deal with non-independent data. The issue motivating these techniques is that when nearby points are placed in different CV folds, spatial autocorrelation among the features at those points makes those folds dependent. This violates the assumptions of the methods, and it could optimistically bias generalization performance estimates if information crosses fold boundaries. Inspired by h-block CV (Burman et al., 1994), there are two general families of spatial CV approaches. Block cross-validation (BLCV) groups geographically proximal points into the same block (Ruß & Brenning, 2010; Roberts et al., 2017). Buffered cross-validation (BFCV) places a buffer between training and validation folds and removes points within the buffer so that they are used in neither set (Le Rest et al., 2014; Pohjankukka et al., 2017). Block size and buffer size are user-defined hyperparameters. One suggested practice is to set them to the median of the spatial autocorrelation ranges of the features. Spatial CV is implemented in some software packages (Brenning, 2012; Muscarella et al., 2014; Valavi et al., 2018) and has been used in previous studies (Zurell et al., 2020; Seo et al., 2021; Valavi et al., 2023) to increase the average distance between training and validation points. One potential issue that may arise with spatial CV techniques is the introduction of covariate shift between folds. By Tobler's first law of geography (Tobler, 1970), things that are close together are more similar than things that are far apart. When creating spatially separated cross-validation folds, it is possible to construct folds with differing feature spaces, and in some cases, this could result in pessimistically biased error estimates. Outside of spatial dependence structure, prior work has also considered the impacts of correlation structure on cross-validation through the lens of latent random effects rather than autocorrelation (Rabinowicz & Rosset, 2020).

There is also prior work on CV procedures that do not assume that the training and test sets are identically distributed. Covariate shift arises frequently in geospatial learning problems when the training feature distribution is different from the feature distribution under which predictions are desired. For example, when considering whether to translocate a threatened species to a currently unoccupied region, natural resource managers may need to construct an SDM with data from a currently occupied region and predict habitat suitability in the new area. Even when predicting at points that interpolate spatially between the training locations, recall that spatial CV approaches for breaking up spatial autocorrelation may actually induce covariate shift. CV algorithms to account for covariate shift have been developed in non-spatial settings, most notably including importance-weighted cross-validation (IWCV, Sugiyama et al. (2007)). IWCV assumes that the training and test data are independent but from different distributions. It develops an asymptotically unbiased estimator by rectifying the loss function with the ratio between the test and training probability densities. IWCV works well when the support of the test distribution is within that of the the training distribution but poorly with minimal overlap between the two, and it does not consider dependent samples. Meyer & Pebesma (2021) suggest an alternative for dealing with covariate shift, by using a dissimilarity index to define an appropriate area of applicability, outside of which cross-validation estimates of model performance do not apply.

Importance-weighted CV procedures (both in prior work and our proposed approach) require density ratio estimates. Density ratios can be estimated efficiently by a variety of algorithms, among which Relative unconstrained Least-Squares Importance Fitting (RuLSIF, Yamada et al. (2011)) is outstanding in theory and practice. Its predecessor, unconstrained Least-Squares Importance Fitting (uLSIF, Kanamori et al. (2009)) provides a closed-form solution by solving a system of linear equations and thus solution is always numerical stable. In importance estimation and covariate shift adaptation tasks, uLSIF is empirically more accurate and faster than other competing importance estimation algorithms (Kanamori et al. (2009)). RuLSIF shares several advantages with uLSIF, like an analytical solution, numerical stability, and robustness, but RuLSIF achieves even better non-parametric convergence. For these reasons, we estimated density ratios with RuLSIF in all experiments below.

When users are unaware of the nuances and applicability of these methods, both standard and spatial CV algorithms may be misused. A few recent studies have compared the CV algorithms above. Roberts et al. (2017), Meyer et al. (2019), and Ploton et al. (2020) conclude that spatial CV is less biased than

non-spatial CV. However, Hoffimann et al. (2021) show that blocking cross-validation can overestimate error. Wadoux et al. (2021) consider sampling probability and conclude that for clustered data KFCV underestimates error but spatial CV algorithms severely overestimate it. Such contradictory conclusions may stem from a lack of clarity on the applicability of a variety CV strategies. This paper seeks to further the discussion by examining their utility across different geospatial scenarios. Milà et al. (2022) specifically considered geographical interpolation and extrapolation and developed the Nearest Neighbour Distance Matching (NNDM) LOOCV, and compared it with no-spatial LOOCV and buffered LOOCV. While we investigate various CV methods under the K-fold setting which is more common in practice.

Finally, by way of background, we remind readers of the concept of semivariogram range (or variogram range) to decide spatial dependence and some vocabulary that will be used below. Semivariogram range is a widely used concept to determine spatial dependence in geostatistics. It is defined as the lag distance where semivariance levels off at its maximum, indicating that pairs of observations at least as far apart as the range are no longer spatially dependent (i.e., they become spatially independent) (Cressie, 2015).

## 3 The Geospatial Problem Setting

Geospatial problems pertain to data with both spatial coordinates (locations) and variables measured at those locations (features); here, we formalize this notion. Our notation draws on that used by Hoffimann et al. (2021). A spatial random variable is a stochastic process $Z : \mathfrak{D} \times \Omega \to \mathbb{R}$, where $\mathfrak{D} \subset \mathbb{R}^d$ is a spatial domain, and $\Omega$ is a sample space. For example, an environmental feature vector $\mathbf{x_i} = (x_i^{(1)}, \ldots, x_i^{(m)})$ is a collection of $m$ processes (e.g., temperature, precipitation) at a specific location $i \in \mathfrak{D}$. Once we obtain feature values of $\mathbf{x_i}$ from an observation, a realization of the processes is produced by fixing $\omega \in \Omega$. In geostatistical learning, we seek to predict the response variable in the test set $\mathbf{y_{te}} = \{y_j\}_{j=1}^{n_{te}}$, given training data $\{X_{tr}, \mathbf{y_{tr}}\} = \{\mathbf{x_i}, y_i\}_{i=1}^{n_{tr}}$ collected from locations in spatial domain $\mathfrak{D}_{tr}$ and test set input features $X_{te} = \{\mathbf{x_j}\}_{j=1}^{n_{te}}$ collected from locations in spatial domain $\mathfrak{D}_{te}$. Assuming the underlying spatial processes are stationary, $\{\mathbf{x_i}\}_{i=1}^{n_{tr}}$ are from the distribution $P_{X_{tr}}$ and are spatially autocorrelated. We use $P$ and $p$ for distributions and probability density functions, respectively. Similarly, $\{\mathbf{x_j}\}_{j=1}^{n_{te}}$ are from $P_{X_{te}}$ and are spatially autocorrelated. If training and test data have the same spatial-temporal extent, then $\mathfrak{D}_{tr}$ and $\mathfrak{D}_{te}$ are considered the same; otherwise, they are different.

Geospatial problems may exhibit different kinds of *data shift*, which happens when the joint distribution of features and responses differs between the training and test sets (Quiñonero-Candela et al., 2009). *Covariate shift*, one type of data shift, occurs when $P_{X_{tr}} \neq P_{X_{te}}$ while $P_{\mathbf{y_{tr}}|X_{tr}} = P_{\mathbf{y_{te}}|X_{te}}$. *Concept shift*, the most intractable form of data shift, happens when the relationship of features to response differs across training and test phases: $P_{\mathbf{y_{tr}}|X_{tr}} \neq P_{\mathbf{y_{te}}|X_{te}}$; we do not consider it in this paper. Another type of data shift is possible but has not been formally defined before: we refer to the case where the training and test sets have different variogram ranges as *range shift*. When $\mathfrak{D}_{tr}$ and $\mathfrak{D}_{te}$ are different, the problem is likely to involve covariate shift and/or range shift.

## 4 A Criterion for Unbiased CV

Rabinowicz & Rosset (2020) prove that when the features of the training and test sets are iid and a latent variable induces correlation structure in the response variable, the CV risk estimate (under squared error loss) is unbiased when the joint distribution of test and training sets is the same as the joint distribution of validation and training folds, for all folds. To extend this result, we consider autocorrelation as the mechanism creating dependence, with potential covariate shift, and a general loss function. With the geospatial settings defined in Sec. 3, we further assume the training and test features have the same domain: $\mathbf{x_i}, \mathbf{x_j} \in \mathscr{X} \subset \mathbb{R}^m$, and the function $f : \mathbf{x} \to y$ is unchanged between training and test sets (i.e., no concept shift). Therefore, the domains of the response variables are also the same: $y_i, y_j \in \mathscr{Y} \subset \mathbb{R}$. Notations with $tr$ and $te$ subscripts (e.g., $T_{tr}$ and $T_{te}$) demonstrate different prediction goals, rather than actual datasets. Specifically, $T_{tr}$ and $T_{te}$ denote training data and test data, respectively; but does not imply that $T$ is the whole dataset. As defined in Sec. 2, $T$ is the training set which is further divided into validation fold $T_k$ and training fold $T_{-k}$. With these assumptions in place, we can state the main theorem.

**Theorem 1.** *If $P_{X_{te}|X_{tr}} = P_{X_k|X_{-k}}, \forall k \in 1, \ldots, K$, then cross-validation is an asymptotically unbiased estimator of the risk $R^{(n)}$.*

*Proof.* Since $p(X_{te}|X_{tr}) = p(X_k|X_{-k})$, we have

$$p(X_{te}, X_{tr})/p(X_{tr}) = p(X_k, X_{-k})/p(X_{-k}). \tag{3}$$

This restatement of the antecedent of the theorem applies to the features $X$, but to evaluate the bias of the risk estimator, we need to work with the full datasets $(T_{tr}, T_{te})$, which include the response variables $y$. To do this, we multiply the LHS by $p(\mathbf{y_{te}}|X_{te})$ and the RHS by $p(\mathbf{y_k}|X_k)$. Since $f$ is assumed constant, these quantities are equal (i.e., $p(\mathbf{y_{te}}|X_{te}) = p(\mathbf{y_{tr}}|X_{tr}) = p(\mathbf{y_k}|X_k)$).

We focus first on the LHS. Multiplying by $p(\mathbf{y_{te}}|X_{te})$ gives

$$\frac{p(X_{te}, X_{tr}) \cdot p(\mathbf{y_{te}}|X_{te})}{p(X_{tr})}.$$

Since $\mathbf{y}$ is conditionally independent of all other variables given its corresponding $X$, we can condition on additional variables (i.e., $p(\mathbf{y_{te}}|X_{te}) = p(\mathbf{y_{te}}|X_{te}, X_{tr})$). Replacing $p(\mathbf{y_{te}}|X_{te})$ with $p(\mathbf{y_{te}}|X_{te}, X_{tr})$ gives

$$\frac{p(X_{te}, X_{tr}) \cdot p(\mathbf{y_{te}}|X_{te}, X_{tr})}{p(X_{tr})}.$$

Next, we multiply the numerator and denominator by $p(\mathbf{y_{tr}}|X_{tr})$, except that by the same reasoning as above, $p(\mathbf{y_{tr}}|X_{tr}) = p(\mathbf{y_{tr}}|X_{tr}, X_{te}, y_{te})$, so we write the multiplier as $\frac{\mathbf{p(y_{tr}}|y_{te}, X_{te}, X_{tr})}{\mathbf{p(y_{tr}}|X_{tr})}$. This gives

$$
\begin{aligned}
&= \frac{p(X_{te}, X_{tr}) \cdot p(\mathbf{y_{te}}|X_{te}, X_{tr}) \cdot p(\mathbf{y_{tr}}|\mathbf{y_{te}}, X_{te}, X_{tr})}{p(X_{tr}) \cdot p(\mathbf{y_{tr}}|X_{tr})} \\
&= \frac{p(\mathbf{y_{te}}, X_{te}, X_{tr}) \cdot p(\mathbf{y_{tr}}|\mathbf{y_{te}}, X_{te}, X_{tr})}{p(\mathbf{y_{tr}}, X_{tr})} \\
&= \frac{p(\mathbf{y_{te}}, X_{te}, \mathbf{y_{tr}}, X_{tr})}{p(\mathbf{y_{tr}}, X_{tr})} \\
&= \frac{p(T_{te}, T_{tr})}{p(T_{tr})}.
\end{aligned}
$$

This quantity, which now includes the response variables as well as features, is $p(T_{te}|T_{tr})$.

The same reasoning applies as we multiply the RHS of Eqn. 3 by $p(\mathbf{y_k}|X_k)$. The condensed process is:

$$
\begin{aligned}
&\frac{p(X_k, X_{-k}) \cdot p(\mathbf{y_k}|X_k)}{p(X_{-k})} \\
&= \frac{p(X_k, X_{-k}) \cdot p(\mathbf{y_k}|X_k, X_{-k}) \cdot p(\mathbf{y_{-k}}|\mathbf{y_k}, X_k, X_{-k})}{p(X_{-k}) \cdot p(\mathbf{y_{-k}}|X_{-k})} \\
&= \frac{p(\mathbf{y_k}, X_k, X_{-k}) \cdot p(\mathbf{y_{-k}}|\mathbf{y_k}, X_k, X_{-k})}{p(\mathbf{y_{-k}}, X_{-k})} \\
&= \frac{p(\mathbf{y_k}, X_k, \mathbf{y_{-k}}, X_{-k})}{p(\mathbf{y_{-k}}, X_{-k})} \\
&= \frac{p(T_k, T_{-k})}{p(T_{-k})} \\
&= p(T_k|T_{-k}).
\end{aligned}
$$

Combining the manipulated LHS and RHS, we can conclude that

$$p(T_{te}|T_{tr}) = p(T_k|T_{-k}).  \qquad (4)$$

Now we are ready to show unbiasedness for the LOO setting. For this setting, Eqn. 4 is written as $p(T_j|T) = p(T_i|T_{-i})$, recalling that $T_j$ is a single intended test instance outside of the full training set $T$, $T_i$ is a single-instance fold, and $T_{-i}$ is the data from the remaining folds. As is typical, we assume that $T_{-i}$ is distributed as $T$ and of size $n$, ignoring the bias from the different sizes of $T_{-i}$ and $T$; this gives $p(T_j|T) = p(T_j|T_{-i}) = p(T_i|T_{-i})$, which is needed for step (1) below. We use shorthand $\mathcal{L}_i$ for $\mathcal{L}(y_i, \hat{y}_i(\mathbf{x_i}; T_{-i}))$ and $\mathcal{L}_j$ for $\mathcal{L}(y_j, \hat{y}_j(\mathbf{x_j}; T_{-i}))$.

$$
\begin{aligned}
\mathbb{E}_T[\hat{R}_{LOOCV}^{(n)}] &= \frac{1}{n}\sum_{i=1}^n \mathbb{E}_{T_{-i},T_i}[\mathcal{L}_i] \\
&= \frac{1}{n}\sum_{i=1}^n \mathbb{E}_{T_{-i}}\left[\int_{\mathcal{Y}}\int_{\mathcal{X}} p(\mathbf{x_i}, y_i|T_{-i})\mathcal{L}_i d\mathbf{x_i}dy_i\right] \\
&\overset{(1)}{=} \frac{1}{n}\sum_{i=1}^n \mathbb{E}_{T_{-i}}\left[\int_{\mathcal{Y}}\int_{\mathcal{X}} p(\mathbf{x_j}, y_j|T_{-i})\mathcal{L}_j d\mathbf{x_j}dy_j\right] \\
&= \frac{1}{n}\sum_{i=1}^n \mathbb{E}_{T_{-i},T_j}[\mathcal{L}_j] \\
&= \frac{1}{n}\sum_{i=1}^n R^{(n-1)} \overset{n\to\infty}{\approx} R^{(n)}
\end{aligned}
$$

All of these claims also hold for KFCV, with more bookkeeping required to account for varying fold sizes. $\square$

Thm. 1 states that to achieve unbiasedness, the distributional relationship between training and test features must be preserved by the split into cross-validation folds. To see the intuition, consider the example of a multivariate Gaussian distribution over $X_{te}$ and $X_{tr}$, and one over $X_k$ and $X_{-k}$. To meet the condition of Thm. 1, the means, variances, and covariances of the distributions should match. In particular, the off-diagonal elements of the covariance matrices should be the same (i.e., $cov(X_{te}, X_{tr}) = cov(X_k, X_{-k})$). This means that if there is spatial (in)dependence between the test and training sets' features, then there should be spatial (in)dependence between the validation and training folds' features. In addition, note that the mean and variances of $X_{te}$ and $X_k$ should match for all folds.

## 5 Framework for Selecting a CV Method

In this section, we present a framework for assessing the relevant characteristics of geospatial problems and linking them to the choice of a CV procedure. The framework uses two criteria to assess whether spatial dependence and/or covariate shift are at play, which sorts the problem into one of four scenarios (Tab. 1). This discrete categorization facilitates our exploration in this paper of which CV strategy is appropriate for each scenario. However, both properties truly exist on spectra rather than being simply present versus absent, and it may be useful in future work to explore other characterizations.

### 5.1 Characteristics of Spatial Problems

First, we characterize the spatial autocorrelation between training and testing sets. Using ideas from spatial statistics, we compare the nearest distance ($d$) between training and test samples, and the semivariogram range ($r$) of the training features. If $d \geq r$, we treat the training and test sets as displaying spatially independence (right column of Tab. 1); otherwise, they are spatially dependent.

Then, we apply the Cramér-von Mises two-sample test (abbr. Cramér test; Anderson (1962)) to assess covariate shift between training and test features. The Cramér test is a multivariate, distribution-free test with a null hypothesis of no covariate shift; i.e., $H_0$: two samples (e.g., $X_{tr}$ and $X_{te}$) are distributed identically. When the p-value ($p$) is smaller than a predefined significance level $\alpha$, we reject $H_0$ and treat covariate shift as present (bottom row of Tab. 1); otherwise we treat it as absent.

Table 1: Geostatistical scenarios determined by semivariogram range and the Cramér test. We consider spatial (in)dependence between training and testing sets by comparing the nearest distance ($d$) between training and test samples with the semivariogram range ($r$) of the training features; and covariate shift by comparing the p-value ($p$) of Cramér test and the user defined significance level $\alpha$.

|  | Spatial Dependence | Spatial Independence |
|---|---|---|
| No Covariate Shift | Scenario SD: $d < r$ and $p \geq \alpha$ | Scenario SI: $d \geq r$ and $p \geq \alpha$ |
| Covariate Shift | Scenario SD + CS: $d < r$ and $p < \alpha$ | Scenario SI + CS: $d \geq r$ and $p < \alpha$ |

Table 2: Summary of how cross-validation methods approach spatial autocorrelation and covariate shift. The first four methods are discussed above; the last method (IBCV) is proposed in the next section. Note that for BLCV and BFCV, the extent to which spatial dependence is ameliorated and covariate shift introduced is moderated by tuning parameters (block or buffer size) and the pattern of spatial autocorrelation.

| Method | Approach to Spatial Autocorrelation | Approach to Covariate Shift (CS) | Intended Scenario |
|---|---|---|---|
| KFCV (Standard K-Fold) | Preserves spatial relationships of training set when creating folds | None | SD |
| IWCV (Importance-Weighted) | Preserves spatial relationships of training set when creating folds | Density ratio estimates | SD + CS |
| BLCV (BLocking) | Reduces spatial dependence across folds | May introduce CS among folds | SI |
| BFCV (BuFfered) | Removes spatial dependence across folds | May introduce CS among folds | SI |
| IBCV (Importance-Weighted Buffered) | Removes spatial dependence across folds | Density ratio estimates | SI + CS |

## 5.2 Characteristics of CV Algorithms

The data splitting mechanism of a CV procedure affects the spatial dependence between training and validation folds. Random partitioning methods, like standard k-fold cross-validation (KFCV) and importance-weighted cross-validation (IWCV), preserve the spatial relationships in the training set, so if the training samples are spatially autocorrelated, then the training and validation folds will be spatially dependent. Blocking cross-validation (BLCV) reduces spatial dependence across folds, though some dependency may remain among points around the splitting boundaries. Intuitively, with more blocks, more spatial dependence remains. Buffered cross-validation (BFCV) makes training and validation folds spatially independent by choosing a buffer as least as large as the spatial autocorrelation, at the cost of losing some data points to the buffer.

CV algorithms also differ in their ability to deal with covariate shift. IWCV is the only one among them that explicitly addresses covariate shift. The spatial partitioning methods, like BLCV and BFCV, have no mechanism to rectify covariate shift but may actually *introduce* covariate shift when splitting the training set into folds. When spatially disparate folds are imposed over spatially autocorrelated features, the folds may be sampling different parts of feature space. The extent to which this issue arises is a function of the degree and pattern of spatial autocorrelation as well as tuning parameters of the approaches like the buffer or block size. In contrast, random partitioning like KFCV is not likely to introduce covariate shift across the folds because it samples uniformly at random across geographic space, which translates to sampling uniformly at random across feature space, even for autocorrelated features.

The approaches that the CV algorithms take to spatial autocorrelation and covariate shift are summarized in Tab. 2. The last line of the table introduces our inspiration for a new method, which we propose in the next section.

## 6 Proposed Method

---
**Algorithm 1** LOOIBCV
---
**Input**: training set $\{T_i\}_{i=1}^n = \{\mathbf{x_i}, y_i\}_{i=1}^n$ and density ratios $\{w_i\}_{i=1}^n$
**Parameters**: buffer size $r$
**Output**: estimated error $Err$

1: **for** i = 1 to n **do**
2:      Remove data points within the buffer area based on the longitude (long) and the latitude (lat) of $T_i$: $[long - r, long + r, lat - r, lat + r]$.
3:      Fit a model $\hat{f}$ on the remaining data $T_{-i-r}$.
4:      Calculate density ratio weighted loss on the validation fold $T_i$: $Err_i = w_i \cdot \mathcal{L}(y_i, \hat{y}_i(\mathbf{x_i}; T_{-i-r}))$.
5: **end for**
6: **Return** the estimated error: $Err = \frac{1}{n} \sum_{i=1}^n Err_i$.

---

Inspired by scenarios in which training and test features are spatially independent and covariate shift is present (SI + CS), we propose Importance Weighted Buffered Cross-Validation (IBCV; Alg. 1), which inherits the advantages of both BFCV and IWCV. It isolates training and validation folds by adding a buffer area between them and applies density ratio weighting to correct the bias from covariate shift. Modified from the IWCV estimator (Sugiyama et al., 2007), the K-fold variant and the leave-one-out variant of IBCV are

$$\hat{R}_{KIBCV}^{(n)} \equiv \frac{1}{K} \sum_{k=1}^K \frac{1}{n_k} \sum_{i \in k^{th}} \frac{p_{te}(\mathbf{x_i})}{p_{tr}(\mathbf{x_i})} \mathcal{L}(y_i, \hat{y}_i(\mathbf{x_i}; T_{-k-r})),$$

$$\hat{R}_{LOOIBCV}^{(n)} \equiv \frac{1}{n} \sum_{i=1}^n \frac{p_{te}(\mathbf{x_i})}{p_{tr}(\mathbf{x_i})} \mathcal{L}(y_i, \hat{y}_i(\mathbf{x_i}; T_{-i-r})),$$

where $T_{-k-r}$ and $T_{-i-r}$ are the training fold, $n_k$ is the size of the validation fold, i.e., the $k^{th}$ fold, and $\frac{p_{te}(\mathbf{x_i})}{p_{tr}(\mathbf{x_i})}$ is the density ratio weight. The buffer size $r$ is decided by the range of spatial autocorrelation, e.g., the largest semivariogram range of all features. In geostatistics, the variogram/semivariogram range indicates

the threshold distance at which two observations are no longer spatially dependent, i.e., they are spatially independent. With the same settings and assumptions of Thm. 1, we show that IBCV is asymptotically unbiased.

**Proposition 2.** *IBCV is asymptotically unbiased:* $\mathbb{E}_T[\hat{R}_{IBCV}^{(n)}] = R^{(n)}$ *when* $n \to \infty$.

*Proof.* We demonstrate the claim for LOOIBCV; it also valid for KIBCV with more bookkeeping for the folds. Step (1) below holds because $T_{-i-r}$ and $T_i$ are independent. Step (2) holds because $T_{-i-r}$ and $T_j$ are independent. We use shorthand $\mathcal{L}_i$ for $\mathcal{L}(y_i, \hat{y}_i(\mathbf{x_i}; T_{-i-r}))$ and $\mathcal{L}_j$ for $L(y_j, \hat{y}_j(\mathbf{x_j}; T_{-i-r}))$.

$$
\begin{aligned}
&\mathbb{E}_T[\hat{R}_{LOOIBCV}^{(n)}] \\
&= \frac{1}{n} \sum_{i=1}^n \mathbb{E}_{T_{-i-r}, T_i} \left[ \frac{p_{te}(\mathbf{x_i})}{p_{tr}(\mathbf{x_i})} \mathcal{L}_i \right] \\
&\overset{(1)}{=} \frac{1}{n} \sum_{i=1}^n \mathbb{E}_{T_{-i-r}} \left[ \int_{\mathscr{Y}} \int_{\mathscr{X}} \frac{p_{te}(\mathbf{x_i})}{p_{tr}(\mathbf{x_i})} p_{tr}(\mathbf{x_i}, y_i) \mathcal{L}_i d\mathbf{x_i} dy_i \right] \\
&= \frac{1}{n} \sum_{i=1}^n \mathbb{E}_{T_{-i-r}} \left[ \int_{\mathscr{Y}} \int_{\mathscr{X}} p_{te}(\mathbf{x_j}) p_{te}(y_j|\mathbf{x_j}) \mathcal{L}_j d\mathbf{x_j} dy_j \right] \\
&\overset{(2)}{=} \frac{1}{n} \sum_{i=1}^n \mathbb{E}_{T_{-i-r}, T_j} \left[ \mathcal{L}_j \right] \\
&= \frac{1}{n} \sum_{i=1}^n R^{(n-1-n_r)} \overset{n \to \infty}{\longrightarrow} \approx R^{(n)}
\end{aligned}
$$

$\square$

Potential users of IBCV should weigh a few caveats. First, IBCV may not be suitable for small datasets. Removing buffer points will have a stronger effect for smaller datasets, potentially leading to a pessimistic bias. Second, IBCV should be expected to struggle with severe covariate shift just as other importance-weighed methods do. Such methods perform poorly when the support of $P_{X_{te}}$ has little overlap with the support of $P_{X_{tr}}$.

# 7 Experiments

In this section, we report on empirical evaluations of our proposed framework and on the performance of several CV estimators on simulated and real datasets. First, we used simulated datasets to estimate the bias of each CV algorithm in terms of risk, following up on Thm. 1. This was possible because we could create multiple simulated landscapes under identical conditions. Second, we created example applications by subsetting from a dataset of bird observations in Oregon, USA to create a diverse set of geospatial scenarios. Finally, to further explore performance of the proposed IBCV algorithm in Scenario SI + CS, we also applied the algorithms to datasets comprising bird observations in Alaska and California housing prices. On these real datasets, it was not possible to estimate risk directly, since we had only one landscape per analysis, so we measured test error as a proxy. To this end, the real datasets were each divided into a training set and a test set, each with known values for the features and response variables. The ground truth test error was the classification error rate or RMSE computed by fitting the model to the training set and applying it to the test set. The estimated test errors were computed via each of the cross-validation procedures on the training set only and compared to the ground truth test error. The subsections below describe each analysis, with more experimental details in the appendix. Code and data are available at `https://github.com/Hutchinson-Lab/Cross-validation-for-Geospatial-Data`.

## 7.1 Simulation Experiments

### 7.1.1 Simulated Data

The simulation settings were inspired by the four main dataset scenarios discussed above (Tab. 1), with subscenarios for some cases and four autocorrelation ranges per subscenario. To construct synthetic landscapes, we simulated two independent, identical, spatial processes $Z_1$ and $Z_2$ with underlying features $\mathbf{x^{(1)}}$

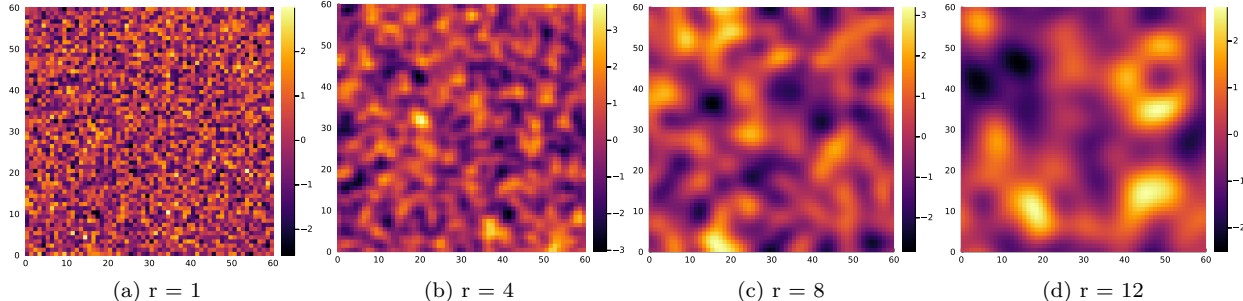

(a) r = 1     (b) r = 4     (c) r = 8     (d) r = 12

Figure 1: Simulated spatial processes (such as those used for $Z_1$ and $Z_2$) with mean=0, variance=1, and four variogram ranges $r$: (a) no, (b) mild, (c) moderate, and (d) strong spatial autocorrelation.

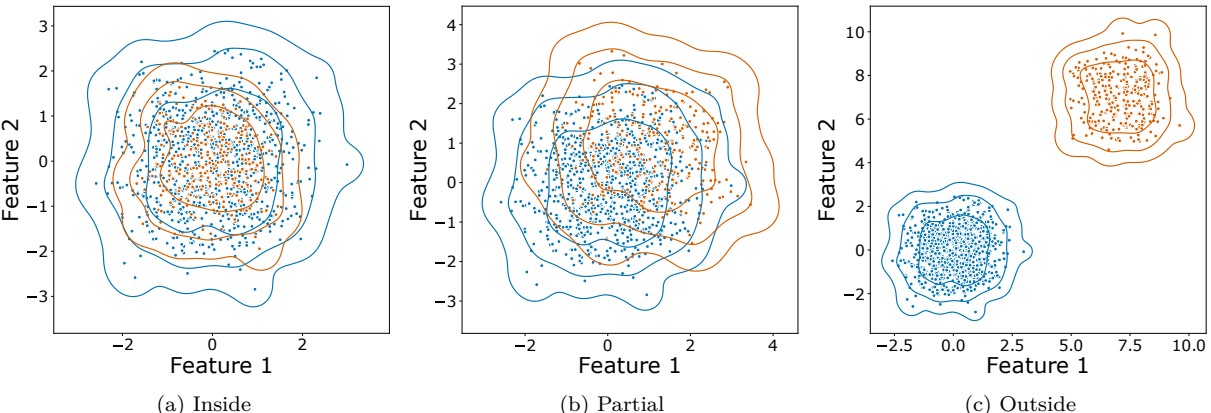

(a) Inside     (b) Partial     (c) Outside

Figure 2: Joint probability density functions of the features in Scenarios SI + CSi (a) and SI + CSp (b). Cases like (c) are beyond the scope of this paper. These examples use $r = 1$ (i.e., no spatial autocorrelation). Blue and orange points denote training and test set feature values, respectively. The inner, middle, and outer contours illustrate the 68%, 95%, and 99.7% quantiles.

and $\mathbf{x^{(2)}}$ on a $60 \times 60$ grid (Fig. 1). We defined a spatial random process by a mean function $m(.)$ and a variogram function $\gamma(h)$. Three parameters are used to describe variograms: the *sill*(s) is the total variance; the *nugget*(n) is the small-scale variability of the data; the *range*(r) is the lag or distance where the variogram levels off to the sill. Pairs of points at least as far apart as the range are not considered spatially autocorrelated. In the simulation, we adopted the Gaussian variogram below:

$$\gamma(h) = (s - n)(1 - exp(-h^2/r^2)) + n, \tag{5}$$

where $h = |z_i - z_j|$ is the distance between any pair of two points $z_i, z_j$ sampled from a continuous two-dimensional spatial domain. Tab. 3 specifies the parameters of the spatial processes for each scenario. In Scenario SI + CS, these variants differed based on whether the support of $P_{X_{te}}$ was contained within (Scenario SI + CSi, Fig. 2a) or partially overlapped with (Scenario SI + CSp, Fig. 2b) the support of $P_{X_{tr}}$. We consider extreme covariate shift, such that the test distribution is entirely outside the training distribution (Fig. 2c), beyond the scope of this paper. Below, we treat SI + CSi as the main representative of the SI + CS scenario; we treat SI + CSp and SI + RS as more challenging/specialized cases. The response variable $\mathbf{y}$ was generated as:

$$\mathbf{y} = \mathbf{x^{(1)}} + \mathbf{x^{(2)}} + \mathbf{x^{(1)}} \cdot \mathbf{x^{(2)}} + \epsilon,$$

Table 3: Parameters of the spatial processes $Z_1, Z_2$ in each scenario: spatially dependent (SD), spatially independent (SI), spatially dependent with covariate shift (SD + CS), spatially independent with covariate shift (SI + CS: CSi - inside, CSp - partially overlapped configurations), and spatially independent with range shift (SI + RS). The hyperparameters $s$, $r$ and $n$ refer to Eqn. 5. We set $n = 0$ for all scenarios. Note that $r = 1$ indicates no spatial autocorrelation as the resolution of the $60 \times 60$ landscape is 1 grid cell.

| Intended Scenario | Training Set | Test Set |
|---|---|---|
| SD, SI, SD + CS | m(.) = 0, s = 1, r = 1, 4, 8, 12 | m(.) = 0, s = 1, r = 1, 4, 8, 12 |
| SI + CSi | m(.) = 0, s = 1, r = 1, 4, 8, 12 | m(.) = 0, s = 0.5, r = 1, 4, 8, 12 |
| SI + CSp | m(.) = 0, s = 1, r = 1, 4, 8, 12 | m(.) = 1, s = 1, r = 1, 4, 8, 12 |
| SI + RS | m(.) = 0, s = 1, r = 12 | m(.) = 0, s = 1, r = 1, 4, 8 |

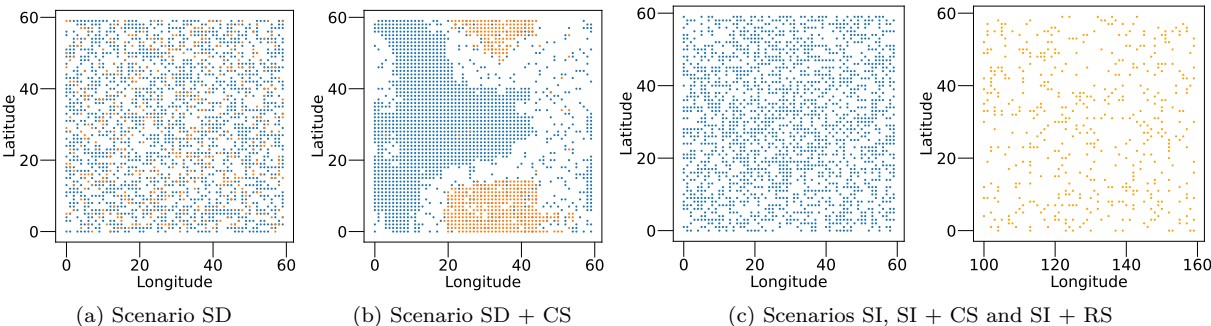

(a) Scenario SD     (b) Scenario SD + CS     (c) Scenarios SI, SI + CS and SI + RS

Figure 3: Examples of spatial sampling in simulations from each scenario. Training points are blue; testing points are orange. Note that (c) shows points far apart in geographic space.

where $\mathbf{x^{(1)}}$ and $\mathbf{x^{(2)}}$ are the (potentially spatially autocorrelated) features, and $\epsilon$ is an i.i.d. Gaussian error term. For every subscenario and range value combination, we repeated this procedure 100 times to allow us to estimate the risk.

Every dataset contained 1800 training and 500 testing samples, but the scenarios varied in the geographic sampling of these grid locations. In Scenario SD, training and test locations were uniformly sampled from the same spatial domain such that test points were interspersed among training points (Fig. 3a). In Scenario SD + CS, they were sampled non-uniformly (e.g., with spatial pattern) from the same spatial domain, so the average distance between training and test points was larger (Fig. 3b). In Scenarios SI, SI + RS and SI + CS, training and test points were uniformly sampled from different spatial domains (Fig. 3c). For the case of SI + RS, the training landscapes had autocorrelation range = 12, while the testing landscapes had different ranges. For SI + CS, the generating distributions changed from training to test landscapes, either concentrating the test features within a smaller region of the support of the training landscapes (CSi) or creating partial overlap between the densities (CSp). Consistent with our proposed framework, in Scenarios SD and SD + CS, the minimum distance between training points and test points is smaller than the variogram range (i.e., $d < r$ in Fig. 3a and Fig. 3b), and in Scenarios SI and SI + CS, the opposite is true (i.e., $d \geq r$ in Fig. 3c). The sampling methods and the underlying Gaussian processes are summarized in Fig. 3 and Tab. 3.

Despite designing these simulations around each scenario in Tab. 1, some sampled datasets did not produce the expected result when testing for covariate shift (Tab. 4). To understand why, the key thing to consider is the role of spatial autocorrelation and its potential to induce covariate shift in samples from these simulation processes. Even when $Z_1$ and $Z_2$ are independent spatial processes, their underlying distributions present stronger spatial pattern as the range increases. With stronger spatial pattern, even a geographically uniform sample from the processes presents a more skewed representation of the feature space. In Scenario SI, where there are two separate realizations of these processes, the likelihood of observing covariate shift between them increases with the range of the underlying spatial autocorrelation. In Scenario SI + RS, range shift

Table 4: Proportion of simulations that reject the Cramér test ($\alpha = 0.01$). We summarize the major characteristics of each scenario based on the above statistical test results from two perspectives: spatially dependent (SD) or spatially independent (SI), and with or without covariate shift (CS). The omitted cases are those that do not fit the scenario definitions (i.e., SD does not occur with $r = 1$ and RS does not occur for $r = 12$ since the training landscape also has $r = 12$.

| Intended Scenario | SD | SI | SD + CS | SI + CSi | SI + CSp | SI + RS |
|---|---|---|---|---|---|---|
| $r = 1$ | 0.00 | 0.00 | - | 1.00 | 1.00 | 0.17 |
| $r = 4$ | 0.01 | 0.00 | 0.86 | 1.00 | 1.00 | 0.03 |
| $r = 8$ | 0.01 | 0.10 | 0.98 | 1.00 | 1.00 | 0.27 |
| $r = 12$ | 0 | 0.33 | 1.00 | 1.00 | 1.00 | - |

exacerbates these differences. In Scenario SD + CS, when spatial sampling is conducted on a single landscape with no spatial autocorrelation ($r = 1$), almost all samples may be identically distributed. However, when spatial sampling is conducted on the distribution with an increasing spatial autocorrelation ($r = 4, 8, 12$), the likelihood of covariate shift increases, since similar feature values are more clustered and more often sampled together (i.e., sampling Fig. 3b on a landscape like Fig. 1(d)). Minor discrepancies in Tab. 4 (i.e., rejecting 1% of simulations in some SD scenarios) are simply a result of the random sampling and the statistical testing framework. In the results below, we excluded all simulations that were not sorted as intended by the Cramér test, to allow appropriate calculations of risk.

### 7.1.2 Simulation Experiment Design

Our primary goal was to measure the bias of each CV method in each scenario, to inform recommendations about which approach to recommend for different types of datasets. Toward this end, we defined the following model class for all simulation experiments:

$$\hat{\mathbf{y}} = \hat{\beta}_0 + \hat{\beta}_1 \mathbf{x^{(1)}} + \hat{\beta}_2 \mathbf{x^{(2)}}.$$

We chose this linear model, which is missing the interaction term of the data-generating model, to mimic the usual case of model misspecification in real applications. We measured performance with root-mean-squared error (RMSE). To evaluate the framework and CV methods, we estimated the risk (Eqn. 2) by averaging the test errors of these models across the simulations in each subscenario. The bias of each CV algorithm is the difference between the average CV estimate (over the simulations) and the estimated risk. The exception to this is Scenario SD + CS. Since the distributions of the training and test features change from one simulation to the next, we are not able to compute the expectations necessary to the target risk by averaging across simulations. In this case, we used the absolute bias as a proxy, defined as the average absolute difference between the CV estimates and the test error on a per-simulation basis: $\frac{1}{n} \sum_{i=1}^{n} |CVest._i - testerror_i|$, where $n$ is the number of simulations sorted into the scenario.

We also compared methods focusing on test error instead of risk, for two reasons. First, even though cross-validation actually estimates risk, practitioners are often interested in test error, given that it conditions on the training set at hand. Second, in the real data analyses below, we can calculate test error but not the true empirical risk, since again we have only a single dataset rather than the simulated replicates. Looking at both in simulated studies may provide insight into the differences between these viewpoints.

Some of the CV methods had hyperparameters to set. We set $k = 9$ folds for all cross-validation methods. The standard KFCV and importance-weighted IWCV split the training set randomly (Fig. 4a). Blocking BLCV partitioned the training samples into blocks and then put these blocks into 9 folds (Fig. 4b). We set the block size of BLCV $= 2, 4, 8, 12$ grids respectively when $r = 1, 4, 8, 12$, such that the block size would mimic the spatial autocorrelation range. This is consistent with the motivation of BLCV, which is to break up spatial autocorrelation. Note that BLCV is sensitive to the block size, and different settings of this hyperparameter produce different results (Fig. 9 in Appendix). In particular, in the presence of spatial autocorrelation, the average BLCV estimate of test error tends to increase with increasing block size, as inter-block covariate shift increases (Fig. 9 in Appendix). It is not trivial in general to set block

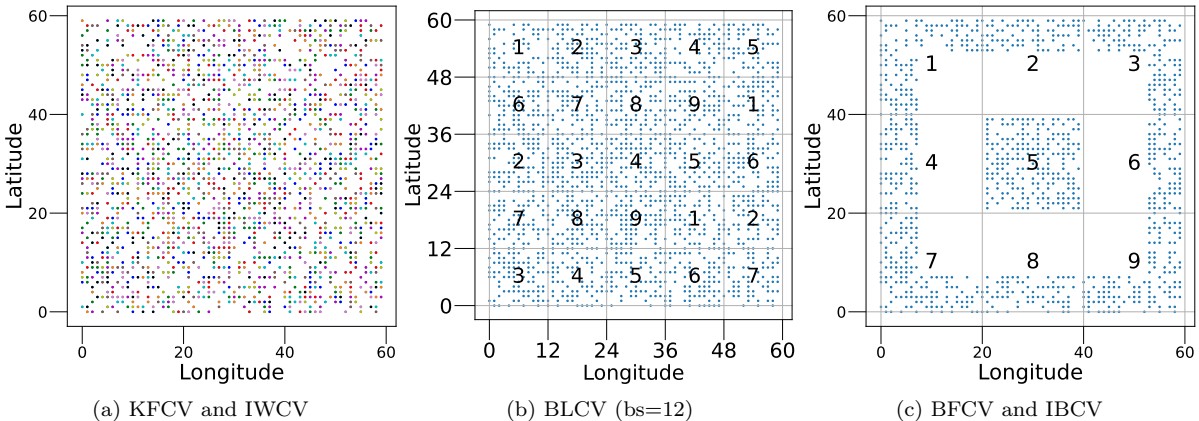

(a) KFCV and IWCV       (b) BLCV (bs=12)       (c) BFCV and IBCV

Figure 4: Visualization of how different CV methods split 1800 training data points on a $60 \times 60$ landscape into 9 folds. (a) Each color represents a fold. (b) An example of BLCV with block size of 12 grid cells, each assigned to one of the 9 folds. We used block sizes of 2, 4, 8, and 12. (c) BFCV and IBCV were based on a grid with 20x20 cells. This example shows fold 5 as the validation fold, where training samples in its surrounding buffer region (buffer size = 12 here) have been removed.

size optimally (Valavi et al., 2018). Buffered BFCV and importance weighted buffered IBCV use a similar grid to BLCV, but with block size of 20, which produces 9 blocks on the $60 \times 60$ landscape, each of which becomes its own fold (Fig. 4c). We removed all training points within a buffer of each validation fold in turn. The size of the buffer was equal to the spatial autocorrelation range of the simulation so that the minimum distance between the training and validation folds was equal or greater than the range, and thus the validation fold could be considered spatially independent. We chose a block size of 20 instead of varying it in order to reduce the amount of data falling into the buffer; with multiple validation folds, the overall buffer area would increase, resulting in fewer training data points. Finally, IWCV and IBCV required density ratio estimates. We applied the Relative unconstrained Least-Squares Importance Fitting (RuLSIF) method to estimate density ratios (Yamada et al., 2011), with $\alpha = 0$ and 50 kernels.

### 7.1.3 Simulation Results

Here, we present results for the four main scenarios: SD, SI, SD+CS, and SI+CSi (denoted SI+CS below). We consider SI+RS and SI+CSp to be auxiliary cases of SI+CS, with SI+RS being somewhat uncommon and SI+CSp being arbitrarily difficult (as we move from SI+CSi through SI+CSp to SI+CSo, the covariate shift becomes increasingly unmanageable). Measurements of bias in the risk estimates of each CV method across the simulated datasets for each scenario are presented graphically in Fig. 5 and numerically in Tab. 5. We examine the results for each scenario below.

In Scenario SD, where $X_{tr}$ and $X_{te}$ are spatially dependent and distributed identically, KFCV is least biased (Tab. 5). Its internal random partitioning mechanism maintains the same spatial relationship between training and validation folds as between training and test sets. The estimates for all methods show increased variation as the range of spatial autocorrelation increases (Fig. 5a). The bias incurred by IWCV reflects estimation error for the density ratios; if the density ratios are set to 1 (as is correct for the SD populations), KFCV and IWCV are equivalent. Similarly, IBCV would be equivalent to BFCV with density ratios of 1, so the discrepancies reflect the density ratio estimation process. The other three spatial CV variants alleviate the spatial dependence between points across fold boundaries; in this case, this is unhelpful because while the folds are decorrelated with each other, the training and test sets are correlated. These methods may also introduce some covariate shift among the folds. In both ways, they overestimate errors.

In Scenario SI, where $X_{tr}$ and $X_{te}$ are spatially independent and distributed identically, BFCV shows promise (Tab. 5, Fig. 5b). Some spatial dependence remains along the boundaries of folds, which may cause underestimation of error, but spatial separation of the folds can also introduce covariate shift, which may cause

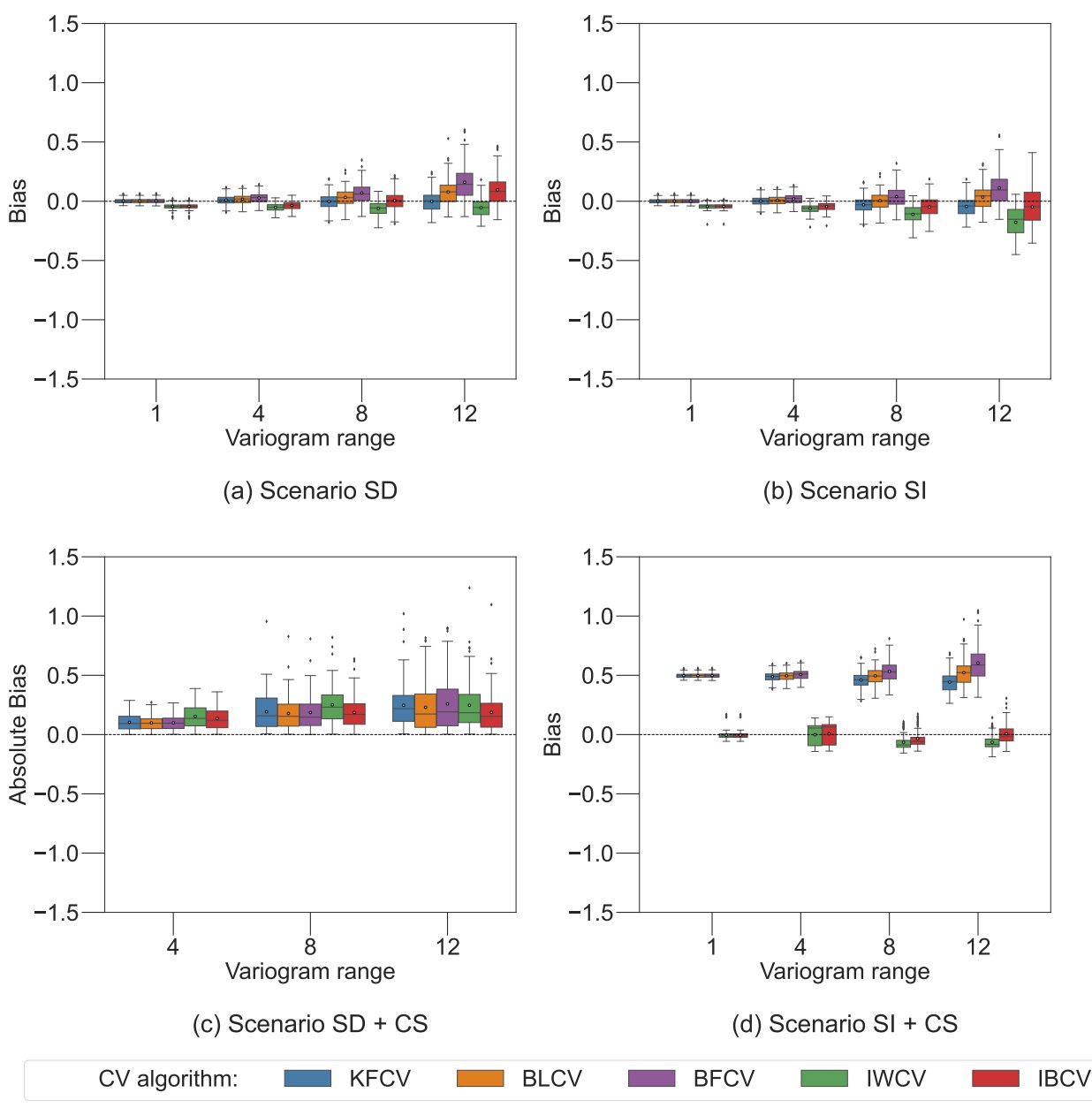

Figure 5: Biases of CV estimates in scenarios with various characteristics: spatially dependent (SD), spatially independent (SI), spatially dependent with covariate shift (SD + CS), and spatially independent with covariate shift (SI + CS). Circles inside the boxes display the mean values of biases. The black dash lines illustrate no bias. (a), (b) and (d) show bias of the average CV estimate to the risk. Since the feature distributions change across simulations in Scenario SD + CS, (c) plots the absolute bias, i.e., the absolute value of CV estimate minus test error in each simulation.

Table 5: Risk perspective: Bias of the average CV estimate (9 folds) with respect to risk, across varying degrees of spatial autocorrelation (i.e., training set range = 1, 4, 8, 12). Bias is calculated as the mean CV estimate minus the risk except for Scenario SD + CS, where bias is calculated as the average of absolute differences between CV estimates and test errors. The least biases in each row of the same scenario block are in bold.

| | KFCV | BLCV | BFCV | IWCV | IBCV | KFCV | BLCV | BFCV | IWCV | IBCV |
|---|---|---|---|---|---|---|---|---|---|---|
| r | | | Scenario SD | | | | | Scenario SI | | |
| 1 | **0.0022** | 0.0023 | 0.0024 | -0.0445 | -0.0443 | **0.0016** | 0.0017 | 0.0018 | -0.0435 | -0.0433 |
| 4 | **0.0103** | 0.0174 | 0.0280 | -0.0517 | -0.0354 | **0.0023** | 0.0094 | 0.0200 | -0.0611 | -0.0449 |
| 8 | **-0.0024** | 0.0030 | 0.0691 | -0.0605 | 0.0062 | -0.0299 | **0.0046** | 0.0388 | -0.1106 | -0.0482 |
| 12 | **-0.0010** | 0.0778 | 0.1602 | -0.0550 | 0.0956 | -0.0438 | **0.0347** | 0.1123 | -0.1778 | -0.0477 |
| | | | Scenario SD + CS | | | | | Scenario SI + CS | | |
| 1 | | | | | | 0.4977 | 0.4978 | 0.4979 | **-0.0052** | **-0.0052** |
| 4 | 0.1043 | 0.1068 | **0.0973** | 0.1651 | 0.1453 | 0.4883 | 0.4954 | 0.5060 | **-0.0018** | 0.0059 |
| 8 | 0.1938 | 0.1861 | **0.1853** | 0.2524 | 0.1872 | 0.4603 | 0.4955 | 0.5320 | -0.0652 | **-0.0347** |
| 12 | 0.2472 | 0.2492 | 0.2583 | 0.2472 | **0.1870** | 0.4429 | 0.5217 | 0.6041 | -0.0646 | **0.0070** |

Table 6: Test error perspective: Proportion of sorted simulations in which each CV algorithm is the closest estimate to the test error, for varying degrees of spatial autocorrelation (i.e., when the range of training set = 1, 4, 8, 12). The highest numbers in each row of the same scenario block are in bold.

| | KFCV | BLCV | BFCV | IWCV | IBCV | KFCV | BLCV | BFCV | IWCV | IBCV |
|---|---|---|---|---|---|---|---|---|---|---|
| r | | | Scenario SD | | | | | Scenario SI | | |
| 1 | 0.23 | 0.20 | **0.25** | 0.11 | 0.21 | **0.25** | 0.24 | 0.20 | 0.16 | 0.15 |
| 4 | 0.20 | 0.07 | **0.35** | 0.22 | 0.15 | 0.17 | 0.11 | **0.36** | 0.25 | 0.11 |
| 8 | 0.23 | 0.21 | 0.15 | **0.26** | 0.14 | 0.12 | 0.10 | **0.42** | 0.23 | 0.12 |
| 12 | **0.37** | 0.17 | 0.03 | 0.30 | 0.13 | 0.19 | 0.10 | **0.39** | 0.28 | 0.03 |
| | | | Scenario SD + CS | | | | | Scenario SI + CS | | |
| 1 | | | | | | 0.00 | 0.00 | 0.00 | **0.57** | 0.43 |
| 4 | 0.16 | 0.09 | **0.45** | 0.08 | 0.21 | 0.00 | 0.00 | 0.00 | **0.55** | 0.45 |
| 8 | 0.17 | 0.09 | 0.31 | 0.07 | **0.36** | 0.00 | 0.00 | 0.00 | 0.19 | **0.81** |
| 12 | 0.10 | 0.15 | 0.25 | 0.15 | **0.35** | 0.01 | 0.00 | 0.00 | 0.32 | **0.67** |

overestimation of error, especially when the range is large. Setting the block size based on the spatial autocorrelation range appears to be a reasonable heuristic in these results, balancing these two potential influences. The upward bias of BFCV increases with range, more substantially than BLCV; this may reflect more covariate shift incurred by BFCV than BLCV and/or the data loss incurred by buffering. Again, the differences between KFCV and IWCV and between BFCV and IBCV are attributable to the density ratio estimates.

In Scenario SD + CS, where $X_{tr}$ and $X_{te}$ are spatially dependent and distributed differently, we used the average of absolute bias to the test error ($|\hat{R}_{CV} - Err_T|$) across simulations to compare the CV algorithms (shown by the Y-axis of Fig. 5(c)). The results omit the case where $r = 1$ (no covariate shift). BFCV performed best with mild and moderate spatial autocorrelation, and IBCV performs best with strong spatial autocorrelation, but the results are similar across all methods (Tab. 5, Fig. 5c).

For Scenario SI + CS, where $X_{tr}$ and $X_{te}$ are spatially independent and distributed differently, the CV algorithms with density ratio weightings have a clear advantage (Tab. 5, Fig. 5d). As may be expected, the improvements of IBCV over IWCV increase with variogram range; i.e., as spatial autocorrelation increases, the importance of the buffering component of the CV method increases as well. IBCV usually performs best on the auxiliary SI+CS cases in the appendix as well, with the caveat that the overall quality of the estimates from all methods decreases with increasing amounts of covariate shift.

Comparing CV methods in terms of test error instead of risk yields slightly different results. Tab. 6 lists the proportion of simulations for which each CV method's estimate was closest to the test error for that simulation (i.e., how often each CV method was best on a per-simulation basis). For Scenario SD, while KFCV was least biased regardless of range, the best method from a test error perspective varied, with BFCV and IBCV sometimes performing better on a larger proportion of simulations. Results from Scenario SI show a similar trend in that KFCV is best for low $r$ while spatial CV approaches perform better as range increases, but the test error analysis favors BFCV while the risk analysis favors BLCV. Results for the scenarios with covariate shift showed more similarity between the two perspectives, in that importance weighting and buffering methods provided best performance.

## 7.2 Example Application

In this section, we provide a real-world example of how different CV methods may affect geospatial analyses. We divided a dataset on bird species distributions in Oregon for training and testing in four distinct ways to set up the different scenarios described above. We were interested to see whether the CV methods that best estimated test performance matched the results from the simulation study above. However, note that in real applications like this, we have only one dataset (in contrast to 100 simulated datasets), so we can only compare estimated and true test errors (Eqn. 1) rather than risks (Eqn. 2). This is more similar to the analysis presented in Tab. 6 than Tab. 5.

### 7.2.1 Oregon 2020 Data

The Oregon 2020 project collected bird species observations by both citizen scientists (a.k.a. community scientists) and professional ornithologists across the state of Oregon by 2020[1] (Robinson et al., 2020). The project was designed to rigorously sample both the geographic space and habitat variety of Oregon. We focused on two common species in the Oregon 2020 dataset: the Hermit Warbler (HEWA) and the Western Tanager (WETA). Our analysis task was to build species distribution models (SDMs), which learn and predict whether a bird species was observed or not across the region of analysis, based on environmental features (Elith & Leathwick, 2009). SDMs are valuable tools for scientists and natural resource managers (Phillips et al., 2004). At each bird observation location, we assembled environmental features to characterize the surrounding habitat. The four features were vegetation indexes computed from remote sensing data using the Tasseled Cap transformation of LandSat imagery (Crist & Cicone, 1984): the mean values of the Tasseled Cap angle (TCA), brightness (TCB), greenness (TCG), and wetness (TCW) in a 600-m radius area around

---

[1]Project website: https://oregon2020.com/

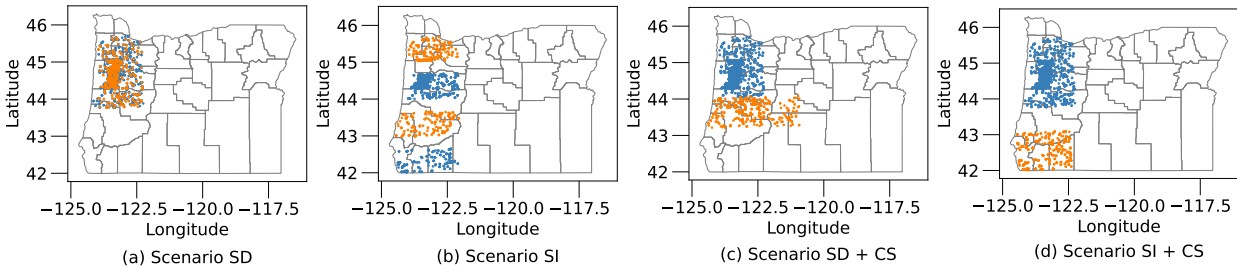

Figure 6: Training and test sets for the four scenarios in the HEWA1800 dataset. Training points are blue and testing points are orange. Subplots (a), (c) and (d) share the same training set.

Table 7: Feature statistics for the Oregon 2020 datasets: Range lists the maximal semivariogram range of training features in degrees. Cramér lists p-value of the Cramér two-sample test whose null hypothesis is that training and test samples are from the same distribution.

| Dataset | Range | Cramér | Dataset | Range | Cramér |
|---|---|---|---|---|---|
| HEWA1000 SD | 0.28 | 0.7123 | HEWA1000 SI | 0.33 | 0.0430 |
| HEWA1800 SD | 0.30 | 0.6184 | HEWA1800 SI | 0.32 | 0.1389 |
| WETA1800 SD | 0.27 | 0.0999 | WETA1800 SI | 0.28 | 0.0649 |
| HEWA1000 SD+CS | 0.28 | 0.0000 | HEWA1000 SI+CS | 0.28 | 0.0000 |
| HEWA1800 SD+CS | 0.30 | 0.0000 | HEWA1800 SI+CS | 0.30 | 0.0000 |
| WETA1800 SD+CS | 0.27 | 0.0000 | WETA1800 SI+CS | 0.27 | 0.0000 |

each survey site. Previous work found these four variables to be predictive of bird species in Oregon (Hopkins et al., 2022).

We created training and testing datasets for three combinations of species and dataset size, each with four geographic layouts to set up the four scenarios of Tab. 1. We assembled datasets with either 1000 or 1800 training observations and in each case tested on 500 held-out observations; datasets are named by their species abbreviation and training sample size (i.e., HEWA1000, HEWA1800 and WETA1800). In all cases, we randomly removed samples from the majority class to produce balanced binary classification tasks, in order to control for any effects of imbalanced data on the classifiers building the SDMs. In Scenarios SD and SD + CS, the training and test samples were spatially proximal (Fig. 6 a and c). In Scenarios SI and SI + CS, the minimum distance between training and test points is larger than the maximum semivariogram range among the four features (Fig. 6 b and d). The p-values of the Cramér test in Scenarios SD and SI for all three datasets are greater than $\alpha = 0.01$, so we did not reject the null hypothesis; the training features were distributed the same as the test features. The p-values are all zeros in Scenarios SD + CS and SI + CS, so we rejected the null hypothesis; covariate shift was present. In summary, each dataset meets the criteria for the intended evaluation scenario.

### 7.2.2 Oregon 2020 Experimental Design

As in the simulation experiments, we sought to determine the best CV approach for each SDM. Whereas in simulation we fit a single regression, we explored five classification models for each SDM: Ridge classifier (Ridge), Linear SVM (LSVM), K-Nearest Neighbors (KNN), Random Forest (RF), and Naive Bayes (NB), and we compared their test errors with the CV error estimates. We used the default hyperparameters from the scikit-learn Python package (Pedregosa et al., 2011) for all models. While peak-to-peak comparison with tuned hyperparameters is critical for comparing state-of-the-art models, an unbiased CV procedure should be able to estimate model performance correctly whether it is high- or low-performing. We evaluated models with classification error rate: the proportion of samples misclassified.

An example showing the splits produced by the CV methods for the HEWA1800 dataset is given in Fig. 7. The CV hyperparameters for HEWA1800 were set as follows; the other analyses followed an analogous

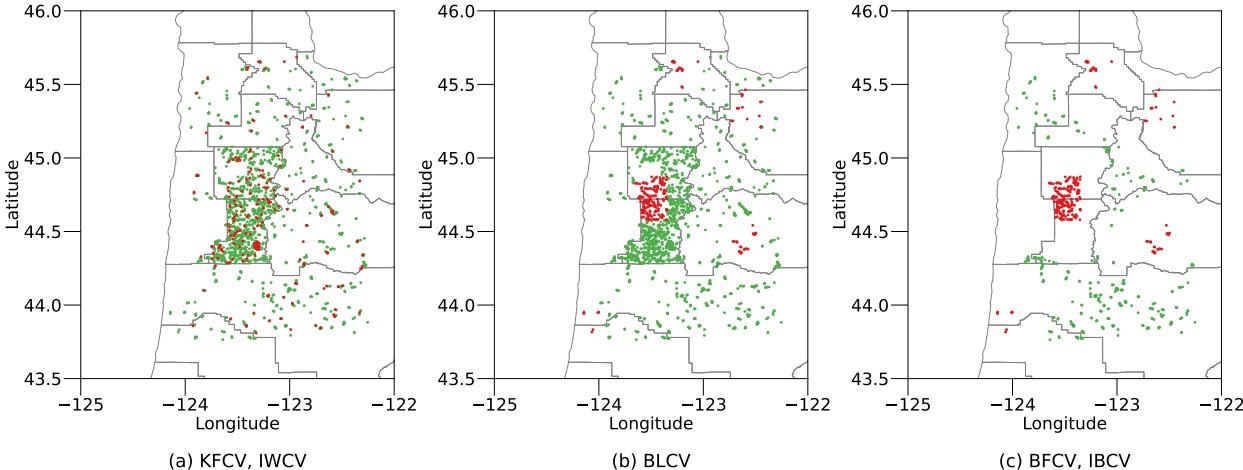

Figure 7: Splitting strategies of five CV algorithms on HEWA1800, Scenario SD dataset when block size = 0.30 degrees: eight training folds (green points) and one validation fold (red points) in each subplot. Compared with (b), training samples in buffer regions have been removed in (c).

procedure. To calculate ranges of the features of training sets, we fitted Matérn variogram functions with the lag class estimated by Scott's rule (Mälicke, 2021). The maximal ranges among the four features are 0.30, 0.32, 0.30 and 0.30 degrees in Scenario SD, SI, SD+CS, SI+CS, respectively (Tab. 7). We set buffer size equal to the maximum range over all features to make training and validation folds spatially independent. We fine-tuned the block size among values equal to the range, double the range, and triple the range, specifically [0.30, 0.60, 0.90] for SD, SD+CS, and SI+CS. For SI, tripling the range produces fewer than 9 folds, so we only selected among [0.32, 0.64]. The Relative unconstrained Least-Squares Importance Fitting (RuLSIF) method was applied to estimate density ratios (Yamada et al., 2011), again with $\alpha = 0$ and the number of kernels = 50.

### 7.2.3 Oregon 2020 Analysis Results

This analysis highlights the importance of the hyperparameters of the CV approaches. In Scenario SD, the test errors generated by the CV methods usually overestimate the true test error (21/25 cases in Tab. 8), and the best method varies across classifiers. In this case, setting the hyperparameters based on the range of the features works well; selecting the best hyperparameters *post hoc* matches this heuristic. In Scenario SI, there is variation in the best CV approach again, and all CV approaches underestimate the true error for all classifiers. Here, setting the hyperparameters for the spatially explicit methods optimally would change the results for KNN and RF, such that BLCV would be the best choice for both. In Scenario SD + CS, all methods underestimate test error for all classifiers when setting the hyperparameters based on feature ranges, and KFCV performs best for four out of five classifiers. With optimal hyperparameter selection, spatially explicit approaches would be better in three of these cases. In Scenario SI + CS, IWCV is the best CV approach for four out of five classifiers, and IBCV for the fifth. With optimal tuning, three of these would shift from IWCV to IBCV. Without hyperparameter tuning, most test errors are underestimated (24/25 cases); with tuning, this is still often the case (11/15 cases).

The results for the other Oregon 2020 datasets (HEWA1000, WETA1800) had similar variability to HEWA1800 (Appx. Tab. 13, Appx. Tab. 14). In all scenarios, test errors for WETA1800 were roughly double those of the HEWA analysis; in these regions, HEWA is more of a habitat specialist (clearly preferring some habitats over others) than WETA (generally widespread), so the environmental features may be less informative overall for the WETA distribution. In Scenario SD, analysis of the other datasets provided less support for spatially explicit CV approaches, and for WETA1800, setting hyperparameters *a priori* versus *post hoc* showed some differences. For Scenario SI, the analyses were more consistent in supporting spatially explicit CV approaches; BLCV and BFCV were usually provided the strongest estimates. Again

Table 8: HEWA1800: model classification test error rates (targets) and 9-fold CV estimates thereof (best estimates in each column in bold). BLCV-best, BFCV-best and IBCV-best estimates are selected from the best ones in the Appx. Tab. 15, for a peak-to-peak comparison. BLCV-range, BFCV-range, and IBCV-range set the tuning parameters *a priori* based on the range of spatial autocorrelation in the features. A dash means that setting the hyperparameters based on the range gives the best value (i.e., the methods are equivalent).

| Model | *Test error (target)* | KFCV | IWCV | BLCV -range | BFCV -range | IBCV -range | BLCV -best | BFCV -best | IBCV -best |
|---|---|---|---|---|---|---|---|---|---|
| | | | | SD | | | | | |
| Ridge | *0.1640* | 0.1844 | 0.1849 | 0.1786 | **0.1780** | 0.1781 | - | - | - |
| LSVM | *0.1640* | 0.1850 | 0.1855 | **0.1752** | 0.1842 | 0.1846 | - | - | - |
| KNN | *0.2020* | 0.2067 | 0.2070 | 0.2084 | 0.2035 | **0.2031** | - | - | - |
| RF | *0.2040* | 0.1972 | 0.1971 | **0.2057** | 0.1945 | 0.1938 | - | - | - |
| NB | *0.1720* | 0.1878 | 0.1884 | 0.1810 | **0.1757** | **0.1757** | - | - | - |
| | | | | SI | | | | | |
| Ridge | *0.2000* | 0.1800 | 0.1797 | 0.1711 | **0.1842** | 0.1839 | - | - | - |
| LSVM | *0.2040* | 0.1828 | 0.1825 | 0.1805 | **0.1989** | 0.1986 | - | - | - |
| KNN | *0.2260* | **0.1861** | 0.1858 | 0.1753 | 0.1697 | 0.1695 | **0.2117** | 0.2028 | 0.2025 |
| RF | *0.2580* | 0.1894 | 0.1892 | 0.1700 | **0.1950** | 0.1947 | **0.2215** | 0.2114 | 0.2111 |
| NB | *0.2140* | 0.1883 | 0.1881 | **0.1892** | 0.1853 | 0.1850 | - | - | - |
| | | | | SD + CS | | | | | |
| Ridge | *0.2140* | **0.1844** | 0.1842 | 0.1786 | 0.1780 | 0.1778 | 0.2083 | **0.2120** | 0.2117 |
| LSVM | *0.2060* | **0.1850** | 0.1848 | 0.1752 | 0.1842 | 0.1840 | 0.2092 | 0.2081 | **0.2078** |
| KNN | *0.2120* | 0.2067 | 0.2064 | **0.2084** | 0.2035 | 0.2032 | - | - | - |
| RF | *0.2000* | **0.1972** | 0.1970 | 0.2057 | 0.1945 | 0.1942 | - | - | - |
| NB | *0.2160* | **0.1878** | 0.1875 | 0.1810 | 0.1757 | 0.1755 | **0.2131** | 0.2114 | 0.2111 |
| | | | | SI + CS | | | | | |
| Ridge | *0.2400* | 0.1844 | **0.2196** | 0.1786 | 0.1780 | 0.2085 | 0.2083 | 0.2120 | **0.2397** |
| LSVM | *0.2460* | 0.1850 | **0.2207** | 0.1752 | 0.1842 | 0.2167 | 0.2111 | 0.2148 | **0.2421** |
| KNN | *0.2420* | 0.2067 | 0.2477 | 0.2084 | 0.2035 | **0.2370** | **0.2439** | 0.2458 | - |
| RF | *0.2380* | 0.1972 | **0.2341** | 0.2057 | 0.1945 | 0.2239 | 0.2438 | 0.2446 | - |
| NB | *0.2660* | 0.1878 | **0.2248** | 0.1810 | 0.1757 | 0.2059 | 0.2131 | 0.2114 | **0.2403** |

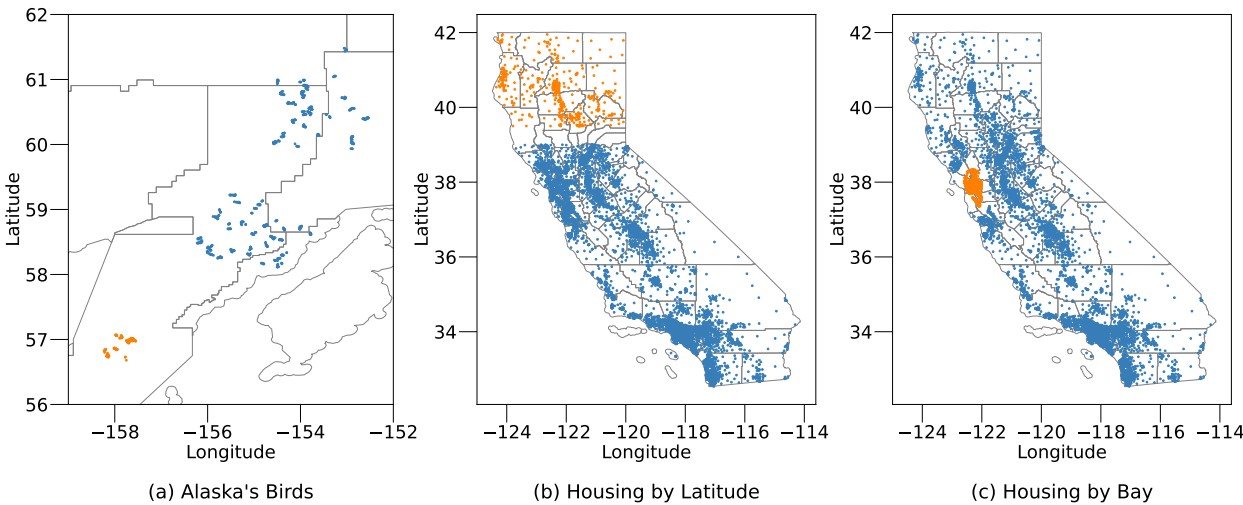

Figure 8: Alaskan National Parks and California Housing dataset spatial layouts. Training points are blue and testing points are orange.

though, the default hyperparameter settings were not the best. The Scenario SD + CS analyses were variable again, with each CV method being the best estimator in at least one case and hyperparameter settings being influential. Finally, in Scenario SI + CS, there was strong support for importance weighting; 10/10 cases were best estimated by IWCV or IBCV. With optimal hyperparameters, IBCV would be best in all cases.

### 7.3 Further Analysis of Scenario SI + CS

We analyzed two more datasets to further explore the performance of IBCV in particular. We performed classification for another species distribution modeling problem using bird data from Alaska, and we explored a regression problem on California housing prices, with two different spatial divisions of training and test points. Both datasets, described below, fell into the SI + CS scenario.

#### 7.3.1 Alaskan National Park Data

This dataset records observations of 15 bird species across three national parks in Alaska, USA[2] (Amundson et al., 2018). These surveys were conducted during May-June, 2004-2006 in Katmai National Park and Preserve (KATM) and Lake Clark National Park and Preserve (LACL), and during May-June, 2008 in Aniakchak National Monument and Preserve (ANIA). We set up a balanced binary classification task for the most prevalent bird species, Golden-crowned Sparrow (GCSP). The features were the proportions of eight land cover classes: water, wetland (wetland vegetation), shrub (shrub vegetation), dshrub (dwarf shrub or herbaceous vegetation), dec (deciduous forest), mixed (mixed deciduous and evergreen forest), spruce (evergreen forest), baresnow (bareground or perennial ice and snow) within a 50-m or 150-m radius circular area around each survey site. The training set consisted of 710 data points (355 non-detections and 355 detections) collected from KATM and LACL in 2004-2006. The test set contained 134 data points (67 non-detections and 67 detections) collected from ANIA in 2008. As shown in Fig 8(c), the nearest distance of training and test samples is 2.01 degrees, greater than the maximal semivariogram range 0.82 degree. The p-value of the Cramér test is 0.

#### 7.3.2 California Housing Data

This dataset[3] (Pace & Barry, 1997) is derived from the 1990 census. The most common task with this dataset is to predict the median house value. We included six numeric features in our models: housing

---

[2]Link to the dataset: https://alaska.usgs.gov/products/data.php?dataid=197

[3]Link to the dataset: https://www.kaggle.com/datasets/camnugent/california-housing-prices

Table 9: SI + CS real datasets: test error (targets) and 10-fold CV estimates thereof (best estimates in each column in bold). BLCV-best, BFCV-best and IBCV-best estimates are selected from the best ones in the Appx. Tab. 15, for a peak-to-peak comparison. BLCV-range, BFCV-range, and IBCV-range set the tuning parameters *a priori* based on the range of spatial autocorrelation in the features. A dash line means setting the tuning parameters based on the range gives the best value (i.e., the methods are equivalent).

| Model | *Test error (target)* | KFCV | IWCV | BLCV -range | BFCV -range | IBCV -range | BLCV -best | BFCV -best | IBCV -best |
|---|---|---|---|---|---|---|---|---|---|
| | | | | Alaskan birds | | | | | |
| Ridge | *0.1866* | 0.3225 | 0.2526 | 0.2777 | 0.2881 | **0.2374** | 0.2696 | 0.2555 | **0.2345** |
| LSVM | *0.1866* | 0.3211 | 0.2488 | 0.2723 | 0.2791 | **0.2244** | 0.2714 | - | - |
| KNN | *0.3284* | 0.3113 | 0.2780 | 0.3030 | **0.3251** | 0.2542 | 0.3279 | - | 0.3254 |
| RF | *0.3657* | 0.3169 | 0.2911 | 0.3176 | **0.3343** | 0.2998 | 0.3460 | **0.3554** | 0.3483 |
| NB | *0.4030* | 0.3521 | 0.2938 | 0.3230 | **0.3597** | 0.2748 | 0.3522 | 0.3663 | **0.3667** |
| | | | | Housing: by latitude | | | | | |
| Linear | *0.5134* | 0.7102 | **0.4298** | 0.7024 | 0.7062 | 0.4286 | - | - | **0.5269** |
| KRR | *0.5931* | 0.7226 | **0.4350** | 0.7135 | 0.7184 | 0.4332 | - | - | **0.5594** |
| SVR | *0.3470* | 0.6496 | **0.4162** | 0.6430 | 0.6564 | 0.4178 | - | - | 0.4650 |
| KNN | *0.5246* | 0.7248 | 0.4483 | 0.7132 | 0.7389 | **0.4558** | - | 0.7370 | **0.5358** |
| RF | *0.4767* | 0.6825 | 0.4274 | 0.6728 | 0.7008 | **0.4318** | - | - | **0.4877** |
| | | | | Housing: by bay | | | | | |
| Linear | *0.7441* | 0.6949 | 0.8121 | 0.6900 | **0.7199** | 0.8518 | 0.7170 | **0.7330** | 0.7793 |
| KRR | *0.7813* | 0.7063 | 0.8349 | 0.7006 | **0.7309** | 0.8722 | 0.7316 | 0.7394 | **0.7988** |
| SVR | *0.6934* | 0.6295 | 0.7310 | 0.6262 | **0.6592** | 0.7744 | 0.6407 | 0.6687 | **0.6838** |
| KNN | *0.7830* | 0.7037 | 0.8232 | 0.6960 | **0.7430** | 0.8821 | 0.7143 | 0.7530 | **0.7859** |
| RF | *0.7176* | 0.6609 | 0.7679 | 0.6504 | **0.6966** | 0.8191 | 0.6702 | **0.7050** | 0.7336 |

median age, total rooms, total bedrooms, population, households, median income. We split the data into training and test sets in two ways: (a) by Latitude, where the test set includes 551 points above 39.5° N and the training set includes 10968 points below 39° N; and (b) by Bay, where the test set consists of 981 points within San Francisco Bay Area and the training set consists of 9775 points whose distance is at least 0.3 degree from any test sample (Fig. 8). For cases (a) and (b), the nearest distances between training and test samples are 0.5 and 0.3 degrees, respectively, which are both greater than their maximal semivariogram ranges of 0.23 and 0.25 degrees, respectively. The p-values of the Cramér tests are zeros for both cases.

### 7.3.3 SI + CS Experiments

For each analysis, we fit five models: Ridge, LSVM, KNN, RF and NB for the birds, and Linear Regression (Linear), Kernel Ridge Regression (KRR), Support Vector Regression (SVR), K-nearest Neighbors (KNN), and Random Forest (RF) for housing prices. We measured classification error rate and RMSE for the birds and prices, respectively. We split the training data with each of the CV methods for each case and measured performance on the held out test sets to serve as the target error for the CV estimators.

The results provide some support for the merits of IBCV, but the importance of hyperparameter selection is evident. Overall, IBCV with the range-based hyperparameter heuristic was best in 4/15 cases, but if optimal hyperparameters could be selected, it would be best in 10/15 cases (Tab. 9). When IBCV is not closest to the target test error, there is still support for both the importance weighting and spatial buffering components of the approach; the other competitive methods are IWCV and BFCV.

## 8 Discussion

Overall, our investigation into cross-validation strategies for geospatial problems suggests that care is warranted when designing evaluation schemes. Across the simulated and empirical data analyses, there is

substantial variation in the results from different CV approaches. For example, the random forest trained on housing data in southern California and tested in northern California was evaluated as 43% above the true test error if using standard KFCV and 9% below the true test error with the proposed IBCV. Inaccurate assessments of model quality may cause problems in a variety of applications, especially when policies are based upon model predictions. These issues propagate into model selection as well; in some cases, the five models being compared had completely different rankings under different CV approaches (e.g., the Alaskan birds dataset under KFCV vs. IBCV).

While important, the task of comparing cross-validation strategies is challenging, particularly due to the distinction between risk and test error. Theoretical analyses of CV algorithms prove that they can produce asymptotically unbiased estimates of risk. Simulation experiments can bear this out, since we can sample landscapes from identical distributions and compute true and estimated risk from these replicated analyses. However, in real data experiments, we can only compute one true test error and one estimate from each CV algorithm. Our simulation results show that the CV estimator that is least biased overall is not always a clear winner in terms of test error (Tab. 5 vs. Tab. 6). This is a fundamental challenge, especially when a modeler has a single training set to analyze and seeks performance estimates that condition on that particular dataset (i.e., test error).

In this work, we explored a discrete framework for sorting analyses into four scenarios. We characterized spatial autocorrelation and covariate shift as present versus absent, by thresholding continuous measures of these attributes. Naturally, this approach has some limitations, and it may be the case that some of the variation in our results comes from varying degrees of spatial autocorrelation and covariate shift among cases within each of the four discrete scenarios. For spatial autocorrelation, we applied a strict distance check to the closest pair of points spanning the training and test sets, but in future work, we will explore a more holistic characterization. For covariate shift, we thresholded the p-value from the Cramér test. In our experience, the Cramér test was very sensitive; in the simulation experiments, many p-values were zero, a few greater than 0.1, and only a small portion in between, so we expect that our results would not change substantially across values of $\alpha$. In practice, for intermediate p-values (e.g., between 0.01 and 0.1), a practitioner might consider cross-validation methods that do and do not address covariate shift. In future work, we will consider more flexible uses of the Cramér test as well as alternative metrics for characterizing covariate shift.

Our proposed algorithm combining the spatial buffering and importance weighting strategies, IBCV, shows some promise for challenging geospatial scenarios. In the simulation experiments where covariate shift was present and the test set was spatially distinct from the training set (SI + CS), IBCV was the best choice of CV method most of the time, both in terms of the bias of the risk estimates and the proportion of cases in which the test error was closest to the truth. The results were more variable on the empirical datasets. IBCV was sometimes the best method with the default hyperparameter settings, but it would have been a more clear favorite if the hyperparameters were set optimally. In 17/20 cases of empirical SI + CS datasets presented above (and 6/10 in the appendix), IBCV yielded better estimates when using hyperparameters that differed from the default settings based on the range of the spatial autocorrelation.

The challenge of hyperparameter tuning was not unique to IBCV. The existing spatial CV methods (BLCV and BFCV) were also often improved by setting hyperparameters to values other than the defaults. Since the optimal hyperparameter results in the experiments above were based on *post hoc* inspection, it is most appropriate to compare methods based on the default hyperparameter settings. The improvements made under the best hyperparameter settings indicate how much potential a method may have, given a robust method for hyperparameter tuning. Currently though, setting these parameters optimally is non-trivial. Nested cross-validation procedures may be an option, but it is undesirable to introduce a hyperparameter search for cross-validation methods that are typically wrapped around other modeling algorithms, which may themselves require hyperparameter tuning. To realize the potential of spatial cross-validation methods, continued research is needed on methods to setting block and buffer sizes appropriately.

Overall, we can make a few recommendations to practitioners based on this study.

- **Scenario SD.** When the training and intended testing datasets are interspersed spatially, random partitioning of the training set (KFCV) is appropriate; there is no covariate shift if both sets are evenly distributed across the landscape, and whatever spatial dependence structure exists between

training and testing data is replicated between CV folds by random partitioning. If a modeler chooses a spatial partitioning instead, it could actually introduce pessimistic bias via covariate shift, especially for long-range autocorrelation.

- **Scenario SI.** When training and test data are geographically separated, spatial partitioning methods like blocking (BLCV) and buffering (BFCV) are good choices, though as discussed above, performance is sensitive to the size of the blocks or buffer.

- **Scenario SD + CS.** The best CV approach was perhaps most variable for this scenario. Importance weighting (IWCV) can be valuable when covariate shift is at play, but spatial partitioning methods sometimes performed better than IWCV. This variation may be driven in part by the relative strengths of the two aspects of the problem (i.e., autocorrelation and covariate shift).

- **Scenario SI + CS.** Our proposed approach of importance-weighted buffered cross-validation (IBCV) can address both of autocorrelation and covariate shift simultaneously. Again, performance was sensitive to the buffer size, and further work on setting this hyperparameter appropriately will help realize the potential of this approach.

Recent papers have come to mixed conclusions and recommendations about the value of spatial CV strategies, and our analysis yields points of both agreement and disagreement with the ongoing discussion. For example, Ploton et al. (2020) argue, with a forest mapping example, that spatial CV should be the norm for ecological and biological studies of spatially autocorrelated processes. Our results do provide evidence for spatial CV strategies in some contexts, though we find that there are some spatial settings (e.g., Scenario SD) where standard k-fold cross-validation is appropriate (#1 above). Wadoux et al. (2021) disagree with Ploton et al. and find instead that spatial CV can be substantially pessimistic in its estimates. Our study finds evidence to support this perspective as well, in particular for Scenario SD. Our agreements with these two disagreeing studies simply highlights one of our main messages: **the appropriate evaluation strategy for a given analysis is context dependent, particularly in relation to the predictive goals of the study.** Another recent paper by Hoffimann et al. (2021) was more similar to ours; those authors also found that spatial blocking CV can produce biased estimates but that standard k-fold CV was also not adequate in the presence of covariate shift. A key difference between their work and ours is that their analysis was limited to analysis of test error, whereas our simulation study looked at bias in both risk estimate and test error estimates. Our multi-faceted analysis provides additional perspective, since test error is more commonly assessed in practice, but theoretical guarantees about CV pertain to risk estimates.

We expect this area of research to be of ongoing importance for applications like species distribution modeling. The ultimate goal of many SDMs is to make useful predictions about species adaptations to changing climatic conditions, including novel "no-analog" climate scenarios, which clearly present covariate shift. Such models sometimes predict that species will move into new spatial areas that they do not currently occupy but that are accessible/adjacent to their current distributions (e.g., shifts pole-ward and/or up in elevation) . Alternately, natural resource managers may consider translocation of species to new spatial areas that are inaccessible to the species by its own dispersal means, as a climate mitigation strategy for threatened taxa. Beyond SDMs, remotely sensed data are enabling the application of machine learning tools to a variety of global monitoring applications (Lacoste et al., 2021), and as these methodologies grow, evaluation methods that are appropriate for spatial prediction tasks are ever more critical. Ongoing work in this field may serve biodiversity conservation and sustainability goals by providing more accurate model assessments for these challenging scenarios.

## 9 Conclusions

This paper has presented an extensive investigation, on both simulated and empirical datasets, into cross-validation algorithms for geospatial applications. In these applications, the presence of spatial autocorrelation and/or covariate shift make it challenging to obtain unbiased estimates of a model's generalization performance. Our theoretical results confirm that in the presence of spatial autocorrelation, the dependence structure between the training and intended test data should match the dependence structure of the CV

folds into which the training data are divided. That is, rather intuitively, the CV folds should be designed to replicate the spatial structure of the intended application. For problems that include both spatial autocorrelation and covariate shift, we developed importance-weighted buffered cross-validation, which both removes data points within a buffer around test folds to break up the spatial dependence and uses density ratios to address covariate shift. IBCV shows promise for challenging spatial problems, though like other spatial CV approaches (based on blocking and buffering), the resulting performance estimates are sensitive to hyperparameters that are not always easy to set properly.

We see several directions for future work. While IBCV shows promise, further developments to aid hyperparameter selection, and/or improvements to the method itself, may be fruitful. The framework developed in Tab. 1 served this investigation adequately, but to make more specific recommendations to practitioners, a more nuanced tool for characterizing spatial autocorrelation and covariate shift (beyond four discrete quadrants) is needed. Finally, we look forward to deploying these methods more broadly, in ecological, economic applications and beyond.

## 10    Acknowledgements

We thank John Kilbride for his work in preprocessing and collecting the remote sensing data for the Oregon bird species distribution experiments, Tom Dietterich for comments on an early version of the manuscript, and three anonymous reviewers for comments that improved the paper. This research was supported in part by the National Science Foundation (NSF) under Grant No. III-2046678 (JW, RAH), the United States Department of Agriculture National Institute of Food and Agriculture (USDA-NIFA) award No. 2021-67021-35344 (AgAID AI Institute; JW, RAH), the National Aeronautics and Space Administration (NASA) under Future Investigators in NASA Earth and Space Science and Technology (FINESST) Grant No. 80NSSC20K1664 (LMH), and the Bob and Phyllis Mace professorship (WDR).

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

# Appendix

## A    Implementation Details

### A.1    Additional Simulation Details

We simulated Gaussian random fields via the fast Fourier transform method (Gutjahr et al., 1997; Hoffimann, 2018). Tab. 3 summarizes the joint distributions of the two spatial features for training and test sets, and the sampling strategies we use to extract training and test data points in each scenario.

### A.2    Density Ratio Estimation Details

We applied the Relative unconstrained Least-Squares Importance Fitting (RuLSIF) method is applied to estimate density ratios (Yamada et al., 2011). Given the definition, we set alpha = 0, and the number of kernels = 50. The algorithm automatically selects an optimal value pair in the range of sigma = $[0.1, 0.5, 1, 2, 3, 4]$ and lambda = $[0.1, 0.3, 0.5, 0.8, 1]$. We fitted Matérn variogram functions with the lag class estimated by Scott's rule (Mälicke, 2021) to calculate ranges of the features of training sets.

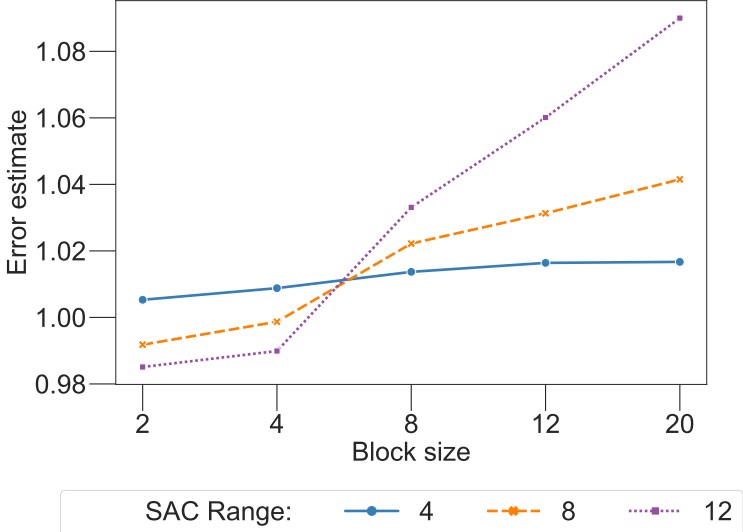

Figure 9: The average BLCV estimates across 100 simulations in Scenario SD, SI, SI + CSi, SI + CSp.

Table 10: Simulation: BLCV estimates with block sizes $= 2, 4, 8, 12, 20$ grids. Best estimates in each column in bold.

|  | Scenario SD | | | Scenario SI | | |
|---|---|---|---|---|---|---|
| r | 4 | 8 | 12 | 4 | 8 | 12 |
| Risk | 0.9912 | 0.9888 | 0.9823 | 0.9994 | 1.0170 | 1.0425 |
| BLCV2 | **1.0053** | **0.9918** | **0.9851** | **1.0053** | 0.9918 | 0.9851 |
| BLCV4 | 1.0088 | 0.9987 | 0.9899 | 1.0088 | 0.9987 | 0.9899 |
| BLCV8 | 1.0137 | 1.0222 | 1.0331 | 1.0137 | **1.0222** | **1.0331** |
| BLCV12 | 1.0164 | 1.0313 | 1.0601 | 1.0164 | 1.0313 | 1.0601 |
| BLCV20 | 1.0167 | 1.0415 | 1.0900 | 1.0167 | 1.0415 | 1.0900 |
|  | Scenario SI + CSi | | | Scenario SI + CSp | | |
| r | 4 | 8 | 12 | 4 | 8 | 12 |
| Risk | 0.5134 | 0.5267 | 0.5384 | 2.0107 | 2.0046 | 1.9702 |
| BLCV2 | **1.0053** | **0.9918** | **0.9851** | 1.0053 | 0.9918 | 0.9851 |
| BLCV4 | 1.0088 | 0.9987 | 0.9899 | 1.0088 | 0.9987 | 0.9899 |
| BLCV8 | 1.0137 | 1.0222 | 1.0331 | 1.0137 | 1.0222 | 1.0331 |
| BLCV12 | 1.0164 | 1.0313 | 1.0601 | 1.0164 | 1.0313 | 1.0601 |
| BLCV20 | 1.0167 | 1.0415 | 1.0900 | **1.0167** | **1.0415** | **1.0900** |

# B   Additional Results

## B.1   Effects of Block Size in Simulation

When there was spatial autocorrelation (i.e., $r \neq 1$ ) in the simulated datasets, we fine-tuned the buffer size $= 2, 4, 8, 12, 20$ grids. In general, the error estimate (the magnitude, not the bias) increased with increasing block size (Fig. 9 and Tab. 10). This finding is consistent with the proposition that covariate shift is increased with larger blocks.

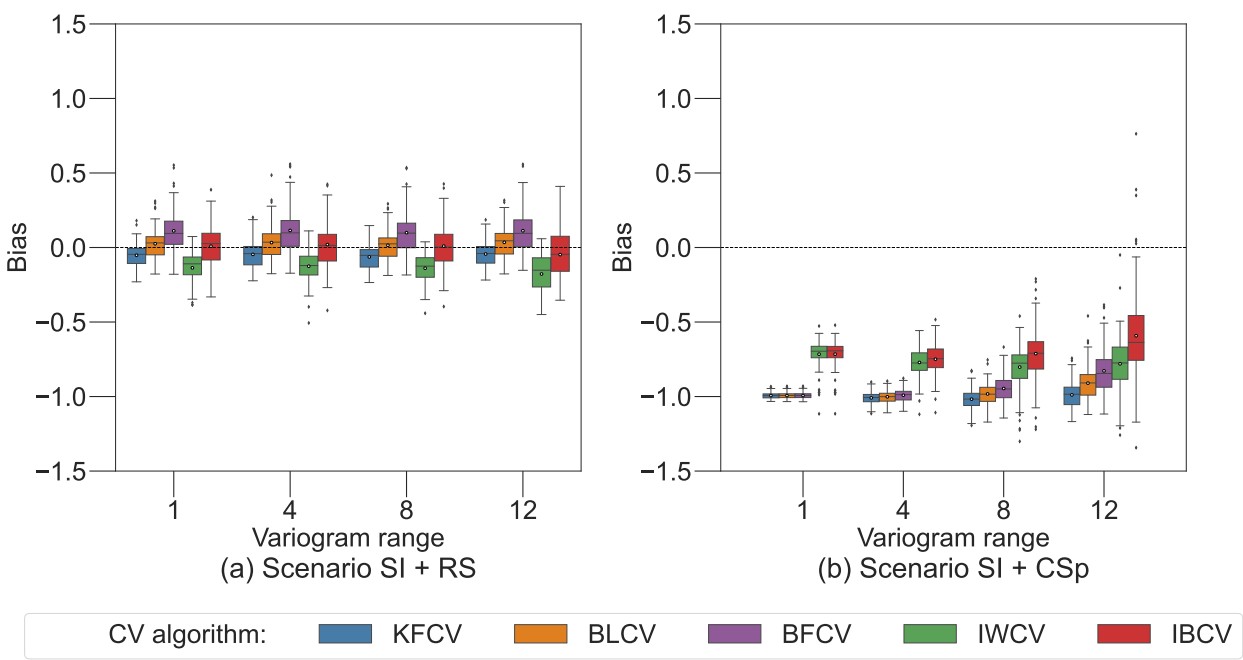

Figure 10: Biases of CV estimates in scenarios with various characteristics: spatially independent with range shift (SI + RS), spatially independent with covariate shift (SI + CSp: partially overlapped). Circles inside the boxes display the mean values of biases. The black dash lines illustrate no bias. Both show bias of the average CV estimate to the risk.

Table 11: Risk perspective: Bias of the average CV estimate (over 9 folds) of risk. The least biased estimates in each row are in bold.

| CV Algo. | KFCV | BLCV | BFCV | IWCV | IBCV | KFCV | BLCV | BFCV | IWCV | IBCV |
|---|---|---|---|---|---|---|---|---|---|---|
| r | | | Scenario SI + RS | | | | | Scenario SI + CSp | | |
| 1 | -0.0523 | 0.0250 | 0.1117 | -0.1365 | **0.0086** | -0.9935 | -0.9934 | -0.9933 | **-0.7157** | **-0.7157** |
| 4 | -0.0463 | 0.0328 | 0.1143 | -0.1256 | **0.0197** | -1.0090 | -1.0019 | -0.9914 | -0.7714 | **-0.7497** |
| 8 | -0.0633 | 0.0151 | 0.0994 | -0.1390 | **0.0093** | -1.0177 | -0.9824 | -0.9460 | -0.8024 | **-0.7121** |
| 12 | | | | | | -0.9889 | -0.9101 | -0.8277 | -0.7805 | **-0.5922** |

## B.2 Special Cases of SI + CS in Simulation Experiments

We treated Scenario SI + RS, where training and test sets are spatially independent with different ranges, and Scenario SI + CSp, where training and test sets are spatially independent and their feature distribution are partially overlapped, as special cases of Scenario SI + CS in our simulation experiments. Results for these scenarios are shown in Fig. 10 and listed in Tabs. 11 and 12.

Table 12: Test error perspective: Proportion of simulations in which a certain CV algorithm is the closest estimate to the test error. Highest numbers in the average row in bold.

| | KFCV | BLCV | BFCV | IWCV | IBCV | KFCV | BLCV | BFCV | IWCV | IBCV |
|---|---|---|---|---|---|---|---|---|---|---|
| r | | | Scenario SI + RS | | | | | Scenario SI + CSp: | | |
| 1 | 0.29 | 0.23 | 0.25 | 0.16 | 0.07 | 0.00 | 0.00 | 0.01 | 0.56 | 0.43 |
| 4 | 0.22 | 0.20 | 0.27 | 0.20 | 0.12 | 0.00 | 0.00 | 0.01 | 0.13 | 0.86 |
| 8 | 0.08 | 0.29 | 0.32 | 0.25 | 0.07 | 0.00 | 0.02 | 0.05 | 0.04 | 0.89 |
| 12 | | | | | | 0.00 | 0.00 | 0.16 | 0.05 | 0.79 |

### B.3  Feature Densities for the Empirical Datasets

Fig. 11, 12, 13, 14, 15, 16 illustrate the distributions of training and test features.

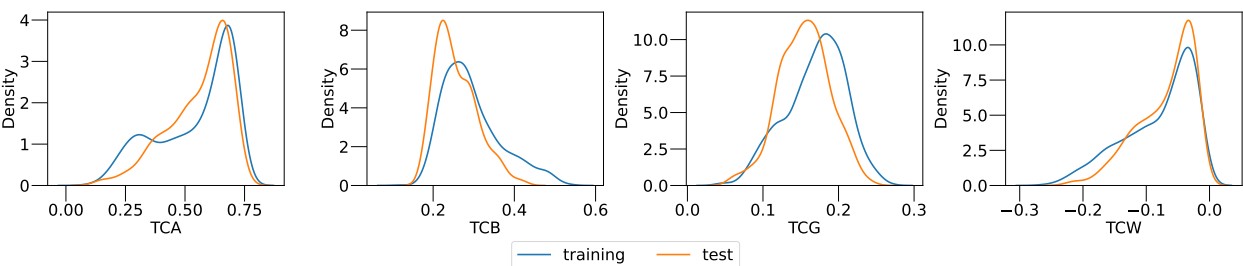

Figure 11: HEWA1000 SI + CS: probability density estimation by kernel smoothing of four features.

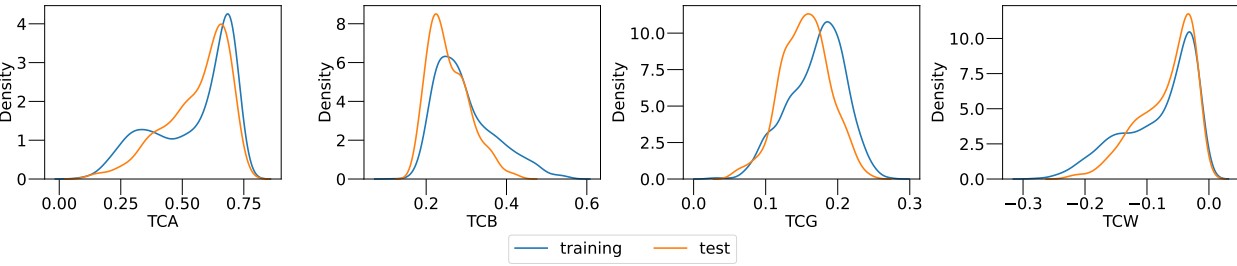

Figure 12: HEWA1800 SI + CS: probability density estimation by kernel smoothing of four features.

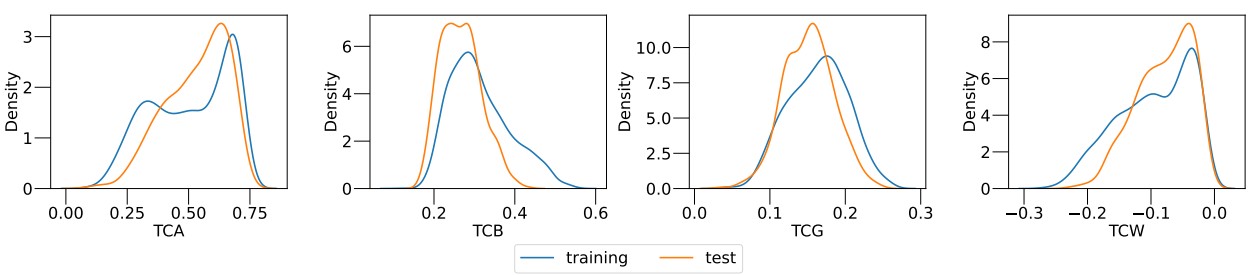

Figure 13: WETA1800 SI + CS: probability density estimation by kernel smoothing of four features.

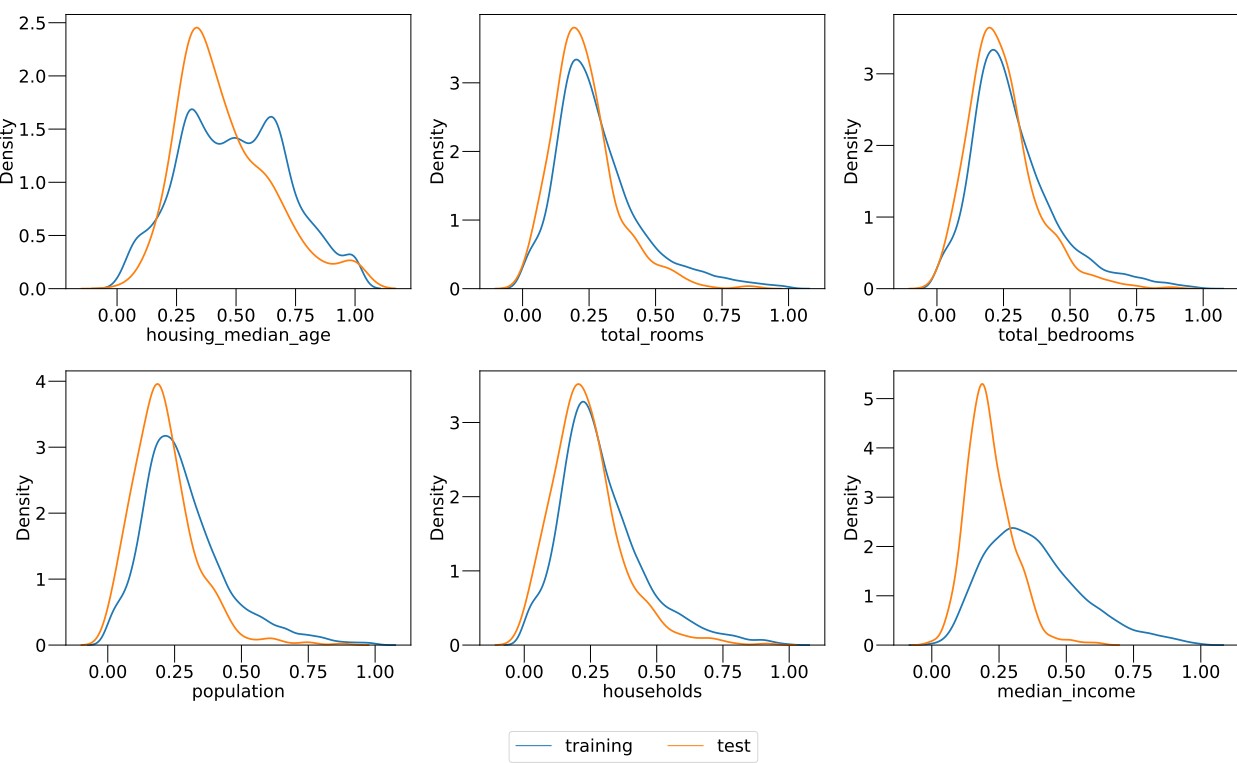

Figure 15: Housing (a) by Latitude: probability density estimation by kernel smoothing of six features.

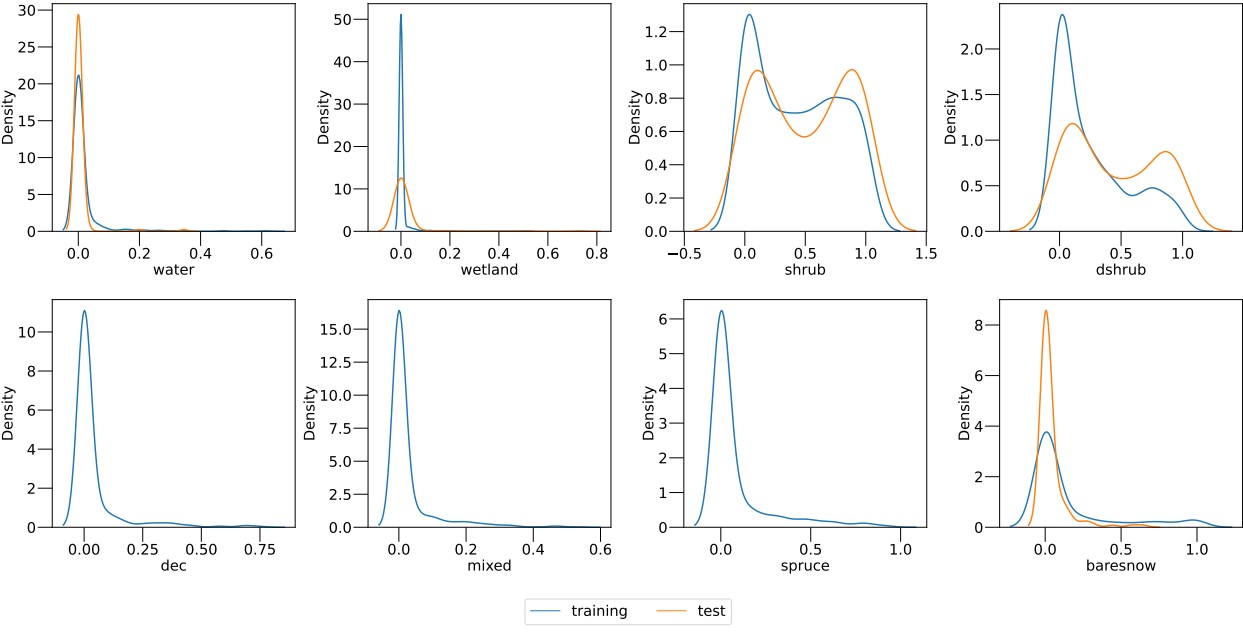

Figure 14: Alaskan birds: probability density estimation by kernel smoothing of eight features. Features dec, mixed and spruce of test samples are all zeros so their density curves are skipped.

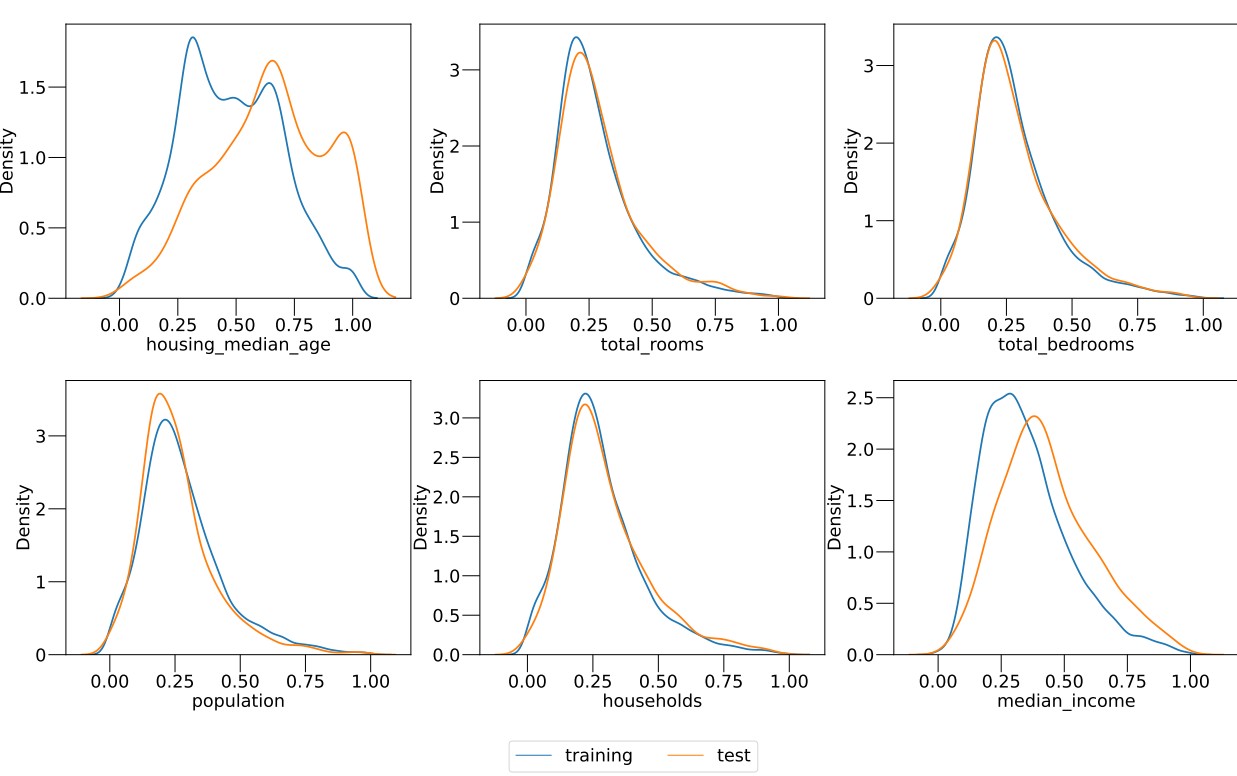

Figure 16: Housing (b) by Bay: probability density estimation by kernel smoothing of six features.

Table 13: HEWA1000: model classification test error rates (targets) and 9-fold CV estimates thereof (best estimates in each column in bold). BLCV-best, BFCV-best and IBCV-best estimates are selected from the best ones in the Appx. Tab. 15, for a peak-to-peak comparison. BLCV-range, BFCV-range, and IBCV-range set the tuning parameters *a priori* based on the range of spatial autocorrelation in the features. A dash means that setting the hyperparameters based on the range gives the best value (i.e., the methods are equivalent).

| Model | *Test error (target)* | KFCV | IWCV | BLCV -range | BFCV -range | IBCV -range | BLCV -best | BFCV -best | IBCV -best |
|---|---|---|---|---|---|---|---|---|---|
| | | | | SD | | | | | |
| Ridge | *0.1700* | 0.1709 | **0.1706** | 0.1664 | 0.1783 | 0.1780 | - | - | - |
| LSVM | *0.1720* | **0.1709** | 0.1706 | 0.1678 | 0.1775 | 0.1773 | - | - | - |
| KNN | *0.1740* | 0.1779 | **0.1777** | 0.1905 | 0.1909 | 0.1906 | - | - | - |
| RF | *0.1740* | 0.1910 | **0.1907** | 0.1964 | 0.1989 | 0.1986 | - | - | - |
| NB | *0.1700* | 0.1729 | 0.1727 | 0.1663 | **0.1678** | 0.1676 | - | - | - |
| | | | | SI | | | | | |
| Ridge | *0.2320* | 0.1890 | 0.1888 | 0.2007 | **0.2173** | 0.2170 | **0.2357** | 0.2411 | 0.2408 |
| LSVM | *0.2280* | 0.1900 | 0.1898 | **0.2047** | 0.2876 | 0.2872 | 0.2406 | 0.2396 | **0.2394** |
| KNN | *0.2440* | 0.2180 | 0.2178 | **0.2292** | 0.2628 | 0.2625 | **0.2402** | 0.2383 | 0.2381 |
| RF | *0.2520* | 0.2120 | 0.2118 | **0.2607** | 0.2692 | 0.2689 | - | 0.2365 | 0.2363 |
| NB | *0.2440* | 0.1950 | 0.1948 | 0.2043 | **0.2110** | 0.2107 | 0.2424 | **0.2441** | 0.2438 |
| | | | | SD + CS | | | | | |
| Ridge | *0.2140* | 0.1709 | **0.2400** | 0.1644 | 0.1783 | 0.2526 | 0.1976 | **0.1997** | - |
| LSVM | *0.2040* | 0.1709 | 0.2430 | 0.1678 | **0.1775** | 0.2489 | 0.2055 | **0.2035** | - |
| KNN | *0.2080* | 0.1779 | 0.2533 | 0.1905 | **0.1909** | 0.2649 | 0.2186 | - | - |
| RF | *0.1840* | **0.1910** | 0.2706 | 0.1964 | 0.1989 | 0.2789 | - | - | - |
| NB | *0.2160* | 0.1729 | 0.2469 | 0.1663 | 0.1678 | **0.2393** | 0.1984 | **0.2037** | - |
| | | | | SI + CS | | | | | |
| Ridge | *0.2420* | 0.1709 | 0.2239 | 0.1644 | 0.1783 | **0.2370** | 0.1976 | 0.1997 | **0.2384** |
| LSVM | *0.2540* | 0.1709 | 0.2245 | 0.1678 | 0.1775 | **0.2334** | 0.2055 | 0.2035 | **0.2544** |
| KNN | *0.2440* | 0.1779 | 0.2370 | 0.1905 | 0.1909 | **0.2484** | **0.2425** | 0.2406 | - |
| RF | *0.2540* | 0.1910 | 0.2507 | 0.1964 | 0.1989 | **0.2572** | 0.2322 | 0.2453 | - |
| NB | *0.2640* | 0.1729 | **0.2289** | 0.1663 | 0.1678 | 0.2222 | 0.1984 | 0.2037 | **0.2546** |

## B.4 Additional Results from the Oregon 2020 Study

Tabs. 13 and 14 show the analagous Oregon 2020 results from the main text, but for the HEWA1000 and WETA1800 datasets, respectively.

Table 14: WETA1800: model classification test error rates (targets) and 9-fold CV estimates thereof (best estimates in each column in bold). BLCV-best, BFCV-best and IBCV-best estimates are selected from the best ones in the Appx. Tab. 15, for a peak-to-peak comparison. BLCV-range, BFCV-range, and IBCV-range set the tuning parameters *a priori* based on the range of spatial autocorrelation in the features. A dash means that setting the hyperparameters based on the range gives the best value (i.e., the methods are equivalent).

| Model | *Test error (target)* | KFCV | IWCV | BLCV -range | BFCV -range | IBCV -range | BLCV -best | BFCV -best | IBCV -best |
|---|---|---|---|---|---|---|---|---|---|
| | | | | SD | | | | | |
| Ridge | *0.3560* | 0.3850 | **0.3828** | 0.3915 | 0.3986 | 0.3963 | 0.3845 | 0.3885 | 0.3863 |
| LSVM | *0.3600* | 0.3822 | **0.3800** | 0.3862 | 0.3896 | 0.3873 | 0.3819 | 0.3799 | **0.3777** |
| KNN | *0.4040* | **0.4028** | 0.4004 | 0.4057 | 0.4094 | 0.4070 | - | - | - |
| RF | *0.4020* | 0.4022 | 0.3999 | **0.4020** | 0.4133 | 0.4109 | - | 0.4105 | 0.4082 |
| NB | *0.3620* | 0.3783 | **0.3762** | 0.3815 | 0.3788 | 0.3766 | - | - | - |
| | | | | SI | | | | | |
| Ridge | *0.4320* | 0.4050 | 0.4043 | **0.4503** | 0.4810 | 0.4802 | 0.4151 | 0.4382 | **0.4375** |
| LSVM | *0.4120* | **0.3894** | 0.3888 | 0.4416 | 0.5029 | 0.5020 | **0.4050** | 0.4226 | 0.4219 |
| KNN | *0.4380* | 0.4011 | 0.4004 | 0.4220 | **0.4279** | 0.4271 | 0.4245 | - | - |
| RF | *0.4660* | 0.3983 | 0.3976 | 0.4205 | **0.4312** | 0.4304 | - | - | - |
| NB | *0.4240* | 0.3872 | 0.3866 | 0.4275 | 0.4268 | **0.4260** | - | - | - |
| | | | | SD + CS | | | | | |
| Ridge | *0.3980* | 0.3850 | 0.3828 | 0.3915 | **0.3986** | 0.3963 | - | - | - |
| LSVM | *0.3960* | 0.3822 | 0.3801 | 0.3862 | **0.3896** | 0.3873 | - | - | - |
| KNN | *0.3680* | 0.4028 | **0.4005** | 0.4057 | 0.4094 | 0.4070 | 0.3846 | - | - |
| RF | *0.3320* | 0.4022 | **0.3999** | 0.4020 | 0.4133 | 0.4109 | 0.3790 | 0.4105 | 0.4082 |
| NB | *0.4020* | 0.3783 | 0.3762 | **0.3815** | 0.3788 | 0.3767 | 0.3923 | **0.3941** | 0.3920 |
| | | | | SI + CS | | | | | |
| Ridge | *0.5100* | 0.3850 | 0.4299 | 0.3915 | 0.3986 | **0.4458** | 0.4253 | 0.4435 | **0.4872** |
| LSVM | *0.5100* | 0.3822 | 0.4255 | 0.3862 | 0.3896 | **0.4278** | 0.4272 | 0.4408 | **0.4803** |
| KNN | *0.4400* | 0.4028 | 0.4410 | 0.4057 | 0.4094 | **0.4398** | - | 0.4199 | - |
| RF | *0.4280* | 0.4022 | **0.4389** | 0.4020 | 0.4133 | 0.4451 | 0.4120 | **0.4212** | - |
| NB | *0.4800* | 0.3783 | **0.4175** | 0.3815 | 0.3788 | 0.4104 | 0.4330 | 0.4318 | **0.4868** |

### B.5 Empirical Results by Block Sizes

Tab. 10, 15, 16, 17 show the CV estimates for each hyperparameter block size setting. We pick the best block size given the target risks or test errors although it is impossible to get these target model errors in real applications. Here we are more interested in the best estimates of each CV algorithm and their peak-to-peak comparisons.

Table 15: Oregon's birds: BLCV, BFCV and IBCV estimates with different block sizes. Best estimates of each CV algorithm in each column in bold.

| Classifier | Ridge | LSVM | KNN | RF | NB |
|---|---|---|---|---|---|
| | Scenario SD, HEWA1000 | | | | |
| Test error | 0.1700 | 0.1720 | 0.1740 | 0.1740 | 0.1700 |
| BLCV0.28 | **0.1644** | **0.1678** | **0.1905** | **0.1964** | **0.1663** |
| BLCV0.56 | 0.1976 | 0.2055 | 0.2425 | 0.2272 | 0.1984 |
| BLCV0.84 | 0.1868 | 0.1867 | 0.2186 | 0.2322 | 0.1808 |
| BFCV0.28 | **0.1783** | **0.1775** | **0.1909** | **0.1989** | **0.1678** |
| BFCV0.56 | 0.1997 | 0.2035 | 0.2406 | 0.2453 | 0.2037 |
| BFCV0.84 | 0.1889 | 0.1902 | 0.2235 | 0.2159 | 0.1848 |
| IBCV0.28 | **0.1780** | **0.1773** | **0.1906** | **0.1986** | **0.1676** |
| IBCV0.56 | 0.1994 | 0.2032 | 0.2403 | 0.2449 | 0.2034 |
| IBCV0.84 | 0.1886 | 0.1899 | 0.2231 | 0.2156 | 0.1845 |
| | Scenario SD, HEWA1800 | | | | |
| Test error | 0.1640 | 0.1640 | 0.2020 | 0.2040 | 0.1720 |
| BLCV0.30 | **0.1786** | **0.1752** | **0.2084** | **0.2057** | **0.1810** |
| BLCV0.60 | 0.2081 | 0.2111 | 0.2439 | 0.2162 | 0.2131 |
| BLCV0.90 | 0.2083 | 0.2092 | 0.2512 | 0.2438 | 0.2098 |
| BFCV0.30 | **0.1780** | **0.1842** | **0.2035** | **0.1945** | **0.1757** |
| BFCV0.60 | 0.2120 | 0.2148 | 0.2494 | 0.2446 | 0.2114 |
| BFCV0.90 | 0.2072 | 0.2081 | 0.2458 | 0.2472 | 0.2084 |
| IBCV0.30 | **0.1781** | **0.1846** | **0.2031** | **0.1938** | **0.1757** |
| IBCV0.60 | 0.2107 | 0.2135 | 0.2478 | 0.2431 | 0.2099 |
| IBCV0.90 | 0.2057 | 0.2066 | 0.2439 | 0.2440 | 0.2070 |
| | Scenario SD, WETA1800 | | | | |
| Test error | 0.3560 | 0.3600 | 0.4040 | 0.4020 | 0.3620 |
| BLCV0.27 | 0.3915 | 0.3862 | **0.4057** | **0.4020** | **0.3815** |
| BLCV0.54 | **0.3845** | **0.3819** | 0.3846 | 0.3790 | 0.3923 |
| BLCV0.81 | 0.4253 | 0.4272 | 0.3915 | 0.4120 | 0.4330 |
| BFCV0.27 | 0.3986 | 0.3896 | **0.4094** | 0.4133 | **0.3788** |
| BFCV0.54 | **0.3885** | **0.3799** | 0.4199 | **0.4105** | 0.3941 |
| BFCV0.81 | 0.4435 | 0.4408 | 0.4160 | 0.4212 | 0.4318 |
| IBCV0.27 | 0.3963 | 0.3873 | **0.4070** | 0.4109 | **0.3766** |
| IBCV0.54 | **0.3863** | **0.3777** | 0.4175 | **0.4082** | 0.3919 |
| IBCV0.81 | 0.4409 | 0.4383 | 0.4136 | 0.4189 | 0.4294 |
| | Scenario SI, HEWA1000 | | | | |
| Test error | 0.2320 | 0.2280 | 0.2440 | 0.2520 | 0.2440 |
| BLCV0.33 | 0.2007 | 0.2047 | 0.2292 | **0.2607** | 0.2043 |
| BLCV0.66 | **0.2357** | **0.2406** | **0.2402** | 0.2343 | **0.2424** |
| BFCV0.33 | 0.2173 | 0.2876 | 0.2628 | 0.2692 | 0.2110 |
| BFCV0.66 | **0.2411** | **0.2396** | **0.2383** | **0.2365** | **0.2441** |
| IBCV0.33 | 0.2170 | 0.2872 | 0.2625 | 0.2689 | 0.2107 |
| IBCV0.66 | **0.2408** | **0.2394** | **0.2381** | **0.2363** | **0.2438** |
| | Scenario SI, HEWA1800 | | | | |
| Test error | 0.2000 | 0.2040 | 0.2260 | 0.2580 | 0.2140 |

| | | | | | |
|---|---|---|---|---|---|
| BLCV0.32 | **0.1711** | **0.1805** | 0.1753 | 0.1700 | **0.1892** |
| BLCV0.64 | 0.2711 | 0.2628 | **0.2117** | **0.2215** | 0.2701 |
| BFCV0.32 | **0.1842** | **0.1989** | 0.1697 | 0.1950 | **0.1853** |
| BFCV0.64 | 0.2733 | 0.2694 | **0.2028** | **0.2114** | 0.2735 |
| IBCV0.32 | **0.1839** | **0.1986** | 0.1695 | 0.1947 | **0.1850** |
| IBCV0.64 | 0.2730 | 0.2691 | **0.2025** | **0.2111** | 0.2732 |
| Scenario SI, WETA1800 | | | | | |
| Test error | 0.4320 | 0.4120 | 0.4380 | 0.4660 | 0.4240 |
| BLCV0.28 | 0.4503 | 0.4416 | 0.4220 | **0.4205** | **0.4275** |
| BLCV0.56 | 0.4851 | 0.4648 | **0.4245** | 0.4135 | 0.4611 |
| BLCV0.84 | **0.4151** | **0.4050** | 0.3691 | 0.3773 | 0.3819 |
| BFCV0.28 | 0.4810 | 0.5029 | **0.4279** | **0.4312** | **0.4268** |
| BFCV0.56 | 0.5221 | 0.4917 | 0.4167 | 0.3953 | 0.4653 |
| BFCV0.84 | **0.4382** | **0.4226** | 0.3836 | 0.3598 | 0.3895 |
| IBCV0.28 | 0.4802 | 0.5020 | **0.4271** | **0.4304** | **0.4260** |
| IBCV0.56 | 0.5212 | 0.4909 | 0.4161 | 0.3946 | 0.4646 |
| IBCV0.84 | **0.4375** | **0.4219** | 0.3829 | 0.3592 | 0.3889 |
| Scenario SD + CS, HEWA1000 | | | | | |
| Test error | 0.2140 | 0.2040 | 0.2080 | 0.1840 | 0.2160 |
| BLCV0.28 | 0.1644 | 0.1678 | 0.1905 | **0.1964** | 0.1663 |
| BLCV0.56 | **0.1976** | **0.2055** | 0.2425 | 0.2272 | **0.1984** |
| BLCV0.84 | 0.1868 | 0.1867 | **0.2186** | 0.2322 | 0.1808 |
| BFCV0.28 | 0.1783 | 0.1775 | **0.1909** | **0.1989** | 0.1678 |
| BFCV0.56 | **0.1997** | **0.2035** | 0.2406 | 0.2453 | **0.2037** |
| BFCV0.84 | 0.1889 | 0.1902 | 0.2235 | 0.2159 | 0.1848 |
| IBCV0.28 | **0.2526** | **0.2489** | **0.2649** | **0.2789** | **0.2393** |
| IBCV0.56 | 0.2724 | 0.2767 | 0.3246 | 0.3327 | 0.2791 |
| IBCV0.84 | 0.2649 | 0.2669 | 0.3085 | 0.2991 | 0.2614 |
| Scenario SD + CS, HEWA1800 | | | | | |
| Test error | 0.2140 | 0.2040 | 0.2080 | 0.1840 | 0.2160 |
| BLCV0.30 | 0.1786 | 0.1752 | **0.2084** | **0.2057** | 0.1810 |
| BLCV0.60 | 0.2081 | 0.2111 | 0.2439 | 0.2162 | **0.2131** |
| BLCV0.90 | **0.2083** | **0.2092** | 0.2512 | 0.2438 | **0.2098** |
| BFCV0.30 | 0.1780 | 0.1842 | **0.2035** | **0.1945** | 0.1757 |
| BFCV0.60 | **0.2120** | 0.2148 | 0.2494 | 0.2446 | **0.2114** |
| BFCV0.90 | 0.2072 | **0.2081** | **0.2458** | 0.2472 | 0.2084 |
| IBCV0.30 | 0.1778 | 0.1840 | **0.2032** | **0.1942** | 0.1755 |
| IBCV0.60 | **0.2117** | 0.2145 | 0.2491 | 0.2443 | **0.2111** |
| IBCV0.90 | 0.2069 | **0.2078** | 0.2454 | 0.2468 | 0.2081 |
| Scenario SD + CS, WETA1800 | | | | | |
| Test error | 0.3980 | 0.3960 | 0.3680 | 0.3320 | 0.4020 |
| BLCV0.27 | **0.3915** | **0.3862** | 0.4057 | 0.4020 | 0.3815 |
| BLCV0.54 | 0.3845 | 0.3819 | **0.3846** | **0.3790** | **0.3923** |
| BLCV0.81 | 0.4253 | 0.4272 | 0.3915 | 0.4120 | 0.4330 |
| BFCV0.27 | **0.3986** | **0.3896** | **0.4094** | 0.4133 | 0.3788 |
| BFCV0.54 | 0.3885 | 0.3799 | 0.4199 | **0.4105** | **0.3941** |
| BFCV0.81 | 0.4435 | 0.4408 | 0.4160 | 0.4212 | 0.4318 |
| IBCV0.27 | **0.3963** | **0.3873** | **0.4070** | 0.4109 | 0.3767 |
| IBCV0.54 | 0.3863 | 0.3778 | 0.4175 | **0.4082** | **0.3920** |
| IBCV0.81 | 0.4410 | 0.4384 | 0.4137 | 0.4189 | 0.4295 |
| Scenario SI + CS, HEWA1000 | | | | | |
| Test error | 0.2420 | 0.2540 | 0.2440 | 0.2540 | 0.2640 |
| BLCV0.28 | 0.1644 | 0.1678 | 0.1905 | 0.1964 | 0.1663 |
| BLCV0.56 | **0.1976** | **0.2055** | **0.2425** | 0.2272 | **0.1984** |

| | | | | | |
|---|---|---|---|---|---|
| BLCV0.84 | 0.1868 | 0.1867 | 0.2186 | **0.2322** | 0.1808 |
| BFCV0.28 | 0.1783 | 0.1775 | 0.1909 | 0.1989 | 0.1678 |
| BFCV0.56 | **0.1997** | **0.2035** | **0.2406** | **0.2453** | **0.2037** |
| BFCV0.84 | 0.1889 | 0.1902 | 0.2235 | 0.2159 | 0.1848 |
| IBCV0.28 | 0.2370 | 0.2334 | **0.2484** | **0.2572** | 0.2222 |
| IBCV0.56 | 0.2494 | **0.2544** | 0.2994 | 0.3093 | **0.2546** |
| IBCV0.84 | **0.2384** | 0.2405 | 0.2818 | 0.2746 | 0.2343 |
| Scenario SI + CS, HEWA1800 | | | | | |
| Test error | 0.2400 | 0.2460 | 0.2420 | 0.2380 | 0.2660 |
| BLCV0.30 | 0.1786 | 0.1752 | 0.2084 | 0.2057 | 0.1810 |
| BLCV0.60 | 0.2081 | **0.2111** | **0.2439** | 0.2162 | **0.2131** |
| BLCV0.90 | **0.2083** | 0.2092 | 0.2512 | **0.2438** | 0.2098 |
| BFCV0.30 | 0.1780 | 0.1842 | 0.2035 | 0.1945 | 0.1757 |
| BFCV0.60 | **0.2120** | **0.2148** | 0.2494 | **0.2446** | **0.2114** |
| BFCV0.90 | 0.2072 | 0.2081 | **0.2458** | 0.2472 | 0.2084 |
| IBCV0.30 | 0.2085 | 0.2167 | **0.2370** | **0.2239** | 0.2059 |
| IBCV0.60 | **0.2397** | **0.2421** | 0.2833 | 0.2734 | 0.2385 |
| IBCV0.90 | 0.2394 | 0.2385 | 0.2799 | 0.2744 | **0.2403** |
| Scenario SI + CS, WETA1800 | | | | | |
| Test error | 0.5100 | 0.5100 | 0.4400 | 0.4280 | 0.4800 |
| BLCV0.27 | 0.3915 | 0.3862 | **0.4057** | 0.4020 | 0.3815 |
| BLCV0.54 | 0.3845 | 0.3819 | 0.3846 | 0.3790 | 0.3923 |
| BLCV0.81 | **0.4253** | **0.4272** | 0.3915 | **0.4120** | **0.4330** |
| BFCV0.27 | 0.3986 | 0.3896 | 0.4094 | 0.4133 | 0.3788 |
| BFCV0.54 | 0.3885 | 0.3799 | **0.4199** | 0.4105 | 0.3941 |
| BFCV0.81 | **0.4435** | **0.4408** | 0.4160 | **0.4212** | **0.4318** |
| IBCV0.27 | 0.4458 | 0.4278 | **0.4398** | **0.4451** | 0.4104 |
| IBCV0.54 | 0.4469 | 0.4304 | 0.4750 | 0.4680 | 0.4510 |
| IBCV0.81 | **0.4872** | **0.4803** | 0.4690 | 0.4756 | **0.4868** |

Table 16: Alaskan birds: BLCV, BFCV and IBCV estimates with block sizes $= 0.82, 0.9, 1.0, 1.1$ degrees. Best estimates of each CV algorithm in each column in bold.

| Classifier | Ridge | LSVM | KNN | RF | NB |
|---|---|---|---|---|---|
| Test error | 0.1866 | 0.1866 | 0.3284 | 0.3657 | 0.4030 |
| BLCV0.82 | 0.2777 | 0.2723 | 0.3030 | 0.3176 | 0.3230 |
| BLCV0.9 | **0.2696** | **0.2714** | 0.2896 | **0.3460** | 0.3193 |
| BLCV1.0 | 0.3119 | 0.3081 | **0.3279** | 0.3252 | **0.3522** |
| BLCV1.1 | 0.2945 | 0.3117 | 0.3216 | 0.3268 | 0.3351 |
| BFCV0.82 | 0.2881 | **0.2791** | **0.3251** | 0.3343 | 0.3597 |
| BFCV0.9 | **0.2555** | 0.3252 | 0.2764 | 0.3424 | 0.3332 |
| BFCV1.0 | 0.3122 | 0.3088 | 0.3582 | **0.3554** | 0.3279 |
| BFCV1.1 | 0.3142 | 0.3464 | 0.3713 | 0.3203 | **0.3663** |
| IBCV0.82 | 0.2374 | **0.2244** | 0.2542 | 0.2998 | 0.2748 |
| IBCV0.9 | 0.2444 | 0.2391 | **0.3254** | **0.3483** | **0.3667** |
| IBCV1.0 | 0.2485 | 0.2594 | 0.3118 | 0.3021 | 0.3031 |
| IBCV1.1 | **0.2345** | 0.2456 | 0.2882 | 0.2800 | 0.2782 |

Table 17: Housing: BLCV, BFCV and IBCV estimates with block sizes $= 0.23, 0.25, 0.5, 1.0, 1.5, 2.0, 2.5$ degrees. Best estimates of each CV algorithm in each column in bold.

| Regressor | Linear | KRR | SVR | KNN | RF |
|---|---|---|---|---|---|
| | (a) By Latitude | | | | |
| Test error | 0.5134 | 0.5931 | 0.3469 | 0.5246 | 0.4771 |
| BLCV0.23 | **0.7024** | **0.7135** | **0.6430** | **0.7132** | **0.6728** |
| BLCV0.5 | 0.7561 | 0.7589 | 0.7049 | 0.7731 | 0.7269 |
| BLCV1.0 | 0.7461 | 0.7643 | 0.6784 | 0.7586 | 0.7185 |
| BLCV1.5 | 0.7938 | 0.8018 | 0.7296 | 0.8151 | 0.8018 |
| BLCV2.0 | 0.7153 | 0.7270 | 0.6671 | 0.7278 | 0.7129 |
| BLCV2.5 | 0.7419 | 0.7656 | 0.6847 | 0.7765 | 0.7557 |
| BFCV0.23 | **0.7062** | **0.7184** | **0.6564** | 0.7389 | **0.7010** |
| BFCV0.5 | 0.7793 | 0.7854 | 0.7285 | 0.8042 | 0.7614 |
| BFCV1.0 | 0.7537 | 0.7741 | 0.6885 | 0.7757 | 0.7346 |
| BFCV1.5 | 0.8062 | 0.8125 | 0.7420 | 0.8344 | 0.8125 |
| BFCV2.0 | 0.7201 | 0.7323 | 0.6714 | **0.7370** | 0.7189 |
| BFCV2.5 | 0.7435 | 0.7671 | 0.6862 | 0.7798 | 0.7557 |
| IBCV0.23 | 0.4286 | 0.4332 | 0.4178 | 0.4558 | 0.4321 |
| IBCV0.5 | 0.5532 | 0.5479 | 0.5504 | 0.5955 | 0.5590 |
| IBCV1.0 | 0.4896 | 0.5094 | **0.4650** | 0.5113 | **0.4877** |
| IBCV1.5 | 0.6938 | 0.6999 | 0.6651 | 0.6777 | 0.6824 |
| IBCV2.0 | **0.5269** | **0.5594** | 0.5284 | **0.5358** | 0.5596 |
| IBCV2.5 | 0.5317 | 0.5584 | 0.5094 | 0.5510 | 0.5471 |
| | (b) By Bay | | | | |
| Test error | 0.7441 | 0.7813 | 0.6933 | 0.7830 | 0.7159 |
| BLCV0.25 | 0.6900 | 0.7006 | 0.6262 | 0.6960 | 0.6510 |
| BLCV0.5 | 0.7053 | 0.7123 | **0.6407** | **0.7143** | **0.6702** |
| BLCV1.0 | 0.6906 | 0.6960 | 0.6139 | 0.6946 | 0.6563 |
| BLCV1.5 | 0.6884 | 0.6980 | 0.6258 | 0.7045 | 0.6640 |
| BLCV2.0 | 0.6618 | 0.6758 | 0.5861 | 0.6670 | 0.6258 |
| BLCV2.5 | **0.7170** | **0.7316** | 0.6216 | 0.6998 | 0.6687 |
| BFCV0.25 | 0.7199 | 0.7309 | 0.6592 | 0.7430 | 0.6970 |
| BFCV0.5 | **0.7330** | **0.7394** | **0.6687** | **0.7530** | **0.7050** |
| BFCV1.0 | 0.6988 | 0.7037 | 0.6239 | 0.7095 | 0.6627 |
| BFCV1.5 | 0.6980 | 0.7095 | 0.6349 | 0.7177 | 0.6749 |
| BFCV2.0 | 0.6676 | 0.6816 | 0.5920 | 0.6754 | 0.6339 |
| BFCV2.5 | 0.7284 | 0.7391 | 0.6350 | 0.7146 | 0.6798 |
| IBCV0.25 | 0.8518 | 0.8722 | 0.7744 | 0.8821 | 0.8196 |
| IBCV0.5 | 0.8529 | 0.8671 | 0.7785 | 0.8790 | 0.8199 |
| IBCV1.0 | 0.7969 | 0.8070 | 0.7076 | 0.8081 | 0.7502 |
| IBCV1.5 | 0.7959 | 0.8161 | 0.7216 | 0.8212 | 0.7678 |
| IBCV2.0 | **0.7793** | **0.7988** | **0.6838** | **0.7859** | **0.7336** |
| IBCV2.5 | 0.8359 | 0.8507 | 0.7191 | 0.8088 | 0.7663 |

