# OpenReview forum: "Cross-validation for Geospatial Data: Estimating Generalization Performance in Geostatistical Problems"
_TMLR — Accepted by TMLR_

### Review · Reviewer_SgZ4 · 2023-06-07

**Summary Of Contributions:**

This paper provides a framework for thinking about how to perform cross validation on spatial regression problems.  Such problems aim to predict a quantity based on a combination of spatial location and feature values observed at that location.  An example application considered is species distribution modeling where one predicts the spatial density of a species (e.g., birds) from point-wise observations.  In these scenarios, observations are spatially correlated (two nearby observations are likely to record similar values) so if observations are dense enough a purely spatial model (e.g., based on Nadaraya-Watson Kernel Regression or K-Nearest Neighbor) could be used; but this is not the goal, the goal is often to understand the relationship between the features and the prediction.  This means if one is to just use standard cross-validation, on NWKR model may do very well without even observing the features.
To address this, the community has proposed a number of methods to split data for cross validation via spatial blocks or with spatial buffers.  This paper attempts to explain and unify these approaches, provide guidance on when to use each, and introduces a new combination of ideas based on both importance weighting and buffering, that works best in some scenarios.

**Audience:**

Yes

**Claims And Evidence:**

Yes

**Requested Changes:**

1.  It is written for a more statistical audience, less for a ML one.  However, similar problems and challenges occur in many subareas of ML.  A few that come to mind (in additional to spatial problems) are learning on temporal data and scientific ML where the parameter space acts like spatial parameters.

2.  The main theorem and derivation is hard to follow, and I think not given in enough detail.  I am particularly thinking about the derivation right after equation (3).  I could not follow all steps.  More steps could be added, and they should each be explained.  If this is the central technical result, it should be explicitly clear.

3.  The discrete decomposition into 4 cases (Table 1) is useful for presentation, but as discussed many other places, these 4 cases are not so clear.  The use of a hypothesis test with user-required parameters seems problematic for making this rigorous.  It is suggested these sorts of tests could be used to algorithmically decide what type of cross-validation is warranted.  I would either (a) note that these tests are only for exposition as they are introduced, or (b) lean into the usefulness of the tests, and argue that they *should* be used for choosing how to perform cross-validation.

4.  The very heavy use of acronyms made the paper hard to read, especially for a reader who had not used similar ones frequently in the past.  I suggest to more times in the paper remind the reader what they stand for.

5.  I took me several reads of the paper to really understand the issue at hand, and how the feature values and spatial values were meant to interact in the prediction.  Not until I looked closely at the bird example did I start to understand this (I think I described it correctly above).  I suggest the authors re-work the introductory settings to describe these issues, and better highlight them in the synthetic experiment.  If the prediction should be purely spatial, then it seems there should be no need to spatial splitting or buffering in cross-validation.  If the prediction should be purely based on features, again it should not matter as long as the spatial coordinates are not used.  It’s only when these should and do interact that this is an interesting challenge.
 (Or if I am totally mis-understanding the issue .. again more clarification, I think, would be helpful).


**Strengths And Weaknesses:**

I think the topic is very important, and I am excited about the goals outlined in the paper.
From my view the literature review is well done, and helps provide useful perspective for the paper.
The attempts at modeling this through simulations and on real data are fairly well done, and support the conclusions and help justify the motivation.

As such I am supportive of accepting this paper in TMLR.

Some weakness are described via the suggestions below.

---

> ### Author Response · Authors · 2023-06-16
> **Thanks for your comments and suggestions**
>
> Your feedback will definitely improve our manuscript. Our response to the requested changes:
> 1. We will take another look through the paper for any terminology or framing that might be adjusted to help bridge gaps between statistical and ML audiences. We agree that learning for temporal data (and spatial-temporal data) and scientific datasets are absolutely related to the ideas in this paper. Since this paper is already long, perhaps we can find a place to mention this connection just briefly, leaving further details to future work.
> 2. It’s helpful to know where we lost you. We will revise this derivation to clarify, adding more steps and explanations. For now, just briefly, since y is decided by its own X, $p(y_{te}|X_{te}) = p(y_{te}|X_{te}, X_{tr})$ and $p(y_{tr}|X_{tr}) = p(y_{tr}|X_{tr}, X_{te}, y_{te})$. In the second line of the three-line equation right below Eqn. (3), we replaced $p(y_{te}|X_{te})$ with $p(y_{te}|X_{te}, X_{tr})$, multiplied the denominator by $p(y_{tr}|X_{tr})$ and multiplied the numerator by $p(y_{tr}|X_{tr}, X_{te}, y_{te})$ at the same time.
> 3. We agree with the weaknesses of this discrete decomposition, as discussed later in the paper. Early in this work, we had hoped that this framework would serve the purpose you mention in (b) of your review; however, as the work progressed, we came to realize that more nuance is likely needed. We will revise the manuscript with your suggestion in (a) in mind, emphasizing early on that these are just for exposition.
> 4. This is helpful to know. We can definitely redefine the acronyms more frequently to help readers.
> 5. You raise some interesting points/questions about the details of the spatial setting that we will work to clarify in the introduction the problem. We think we can highlight the bird SDM use case earlier and more prominently as a motivating/running example. We do need to clarify: We did not include coordinates as features in ML models. In all of the experiments, each data point (i.e. data example) is associated with its geospatial coordinates - BLCV, BFCV, and IBCV split data based on these coordinates. We agree that there are some settings where spatial considerations for cross-validation are not necessary. However, even without spatial coordinates as features, issues can arise when the features are 1) spatially autocorrelated, 2) correlated with each other to some degree, and 3) present covariate shift for geospatial predictions. (Again, we also focus on the usual case where the model cannot capture p(y|x) perfectly.)

---

> > ### Comment · Reviewer_SgZ4 · 2023-06-27
> > **proof of Theorem 4**
> >
> > I agree with the other reviewer.  My main lingering concern is how the writing for proof of Theorem 1 will look in the pdf.  Let us know when it is updated in the pdf, and I will take a closer look.

---

> ### Comment · Reviewer_SgZ4 · 2023-07-11
> **revision**
>
> Thanks for the revision.  It addresses all of my concerns, especially the much more thorough proof of Theorem 1.
>
> I still have two minor requests, however, with regard to that proof.
> 1. In the equation right before equation (4), the analysis makes a big jump from a complex expression to $p(T_k \mid T_{-k})$.  Please elaborate on this derivation.
> 2. After equation (4), the paper replaces the RHS $p(T_{te} \mid T_{tr})$ with $p(T_j \mid T)$ where $T$ is the training set, and $T_j$ is a fold, implying that $T_j \subset T$.  Please explain why it is ok to have the test part $T_{te}$ be a subset $T_{tr}$ or otherwise clarify what is going on here.
>
> thanks!

---

> > ### Author Response · Authors · 2023-07-12
> > **Thank you for another careful review of the proof**
> >
> > 1. We tried to go more in depth for the LHS derivation and leave the RHS condensed, since it follows the same reasoning. Which of the following would be more helpful?
> > - a. Adding another step in the LHS derivation in between that last equation and P(T_te | T_tr).
> > - b. Expanding the RHS derivation to match the LHS derivation.
> > - c. Both.
> >
> > 2. In fact, T_j is not meant to be part of the training set T (this notation is introduced in the Background section). We propose to clarify this by changing “...recalling that T_j is an intended test instance, T is the full training set, …” to “...recalling that T_j is a single intended test instance outside the full training set T,...” Would that be clear?

---

> > > ### Comment · Reviewer_SgZ4 · 2023-07-12
> > > **proof**
> > >
> > > 1.  Option a would be best.
> > >
> > > 2.  Yes, that would make it more clear.  The notation in the context within the proof is a bit confusing, and this would help.

---

### Review · Reviewer_T2Qn · 2023-06-12

**Summary Of Contributions:**

The manuscript provides an overview of challenges in using cross-validation for geospatial problems, characterising different conditions systematically and explaining what kind of algorithms can be used in condition to learn unbiased risk estimates. In addition, the authors provide a new cross-validation variant applicable for one of the cases for which no previous unbiased methods were available. The authors provide comprehensive experimental evaluation of the methods on both synthetic and real data. It is likely that the paper will be interesting for practitioners needing CV for geospatial applications, and further theoretical development building on this is also possible.

**Audience:**

Yes

**Broader Impact Concerns:**

No concerns. The paper makes contributions that reduce the risk of incorrect modelling choices and has only positive impacts.

**Claims And Evidence:**

Yes

**Requested Changes:**

The paper is overall well written and easy to read, but I do not think the length (21.5 pages before references) is properly justified by the content. It would probably be difficult to compress this all the way down to the recommended 12 pages, but something like 16 pages would make the paper more readable and could be achieved by moving some of the experimental material to a supplement and by compressing the verbose presentation style. I admit this is to an extent a personal preference, so this is not to interpreted as direct request for changing the paper.

I have some trouble following Theorem 1 and would like to see it clarified. The main claim sounds reasonable, but it is difficult to follow what exactly happens in the proof since you are multiplying both sides with different constants (that I presume are actually the same under your assumptions) and are quite liberally introducing new terms. You should at least justify what happened on the second and third lines of the three-line equation right below Eq. (3) since especially how the term $p(y_{tr}|X_{te},X_{tr},y_{te})$ appears is not really obvious. Overall, I am having a bit of trouble understanding why the first part of the proof is even needed -- I would have been perfectly happy to start from Eq. (4) as the underlying assumption, and it feels like a more straightforward proof could also be done. If $p(X|X)$ and $p(y|X)$ are both the same in the two cases then quite obviously the same holds for the data sets.

The characterisation of the four different cases in Table 1 is nice and likely to be interesting as a general guide for practitioners, but I find hard thresholding based on a p-value quite dangerous since p-values are not really that reliable indicators. Wouldn't it make more sense for a practitioner to try out both methods when the p-value is close to the threshold? Some discussion on the possible problems of p-value thresholding should be added.

**Strengths And Weaknesses:**

Strengths
- Addresses an important and non-trivial problem and delivers concrete algorithms for practical use
- Very comprehensive experimentation
- Gives concrete recommendations for practitioners

Weaknesses
 - Overly long
 - The new algorithm for the SI+CS scenario is fairly straightforward generalisation of previous methods
 - Paper mentions code is provided in the Supplement, but I could not find it. The value of the paper somewhat depends on the quality of the code, since robust and reliable tools are needed for practitioners to actually use these techniques

---

> ### Author Response · Authors · 2023-06-16
> **Thanks for your comments and suggestions**
>
> Your perspectives and feedback will definitely help us improve our manuscript. Response to the weakness and requested changes:
> 1. We agree that the paper is long, and we are open to condensing it further. We will take another read through and look for more pieces that could move to the supplemental material.
> 2. We agree that IBCV is the straightforward next step in this space. That said, many widely-used cross-validation methods are simple, and despite its simplicity, it has not yet been discussed in the literature. We provide theoretical proof of the unbiasedness property and practical usefulness in a common scenario. We emphasize that IBCV is only one of our paper's contributions; we believe the overview of the CV settings and algorithms is also valuable.
> 3. We had planned to provide code if the paper is accepted, via GitHub. If the reviewers need code in order to make a decision, we can look at options for providing it (while adhering to TMLR’s doubly anonymous review policies).
> 4. It’s very helpful to hear where you had trouble with the proof. We plan to revise this with more clarification to the reader for each step. For now, briefly, since y is decided by its own X, $p(y_{te}|X_{te}) = p(y_{te}|X_{te}, X_{tr})$ and $p(y_{tr}|X_{tr}) = p(y_{tr}|X_{tr}, X_{te}, y_{te})$. In the second line of the three-line equation right below Eqn. (3), we replaced $p(y_{te}|X_{te})$ with $p(y_{te}|X_{te}, X_{tr})$, multiplied the denominator by $p(y_{tr}|X_{tr})$ and multiplied the numerator by $p(y_{tr}|X_{tr}, X_{te}, y_{te})$ at the same time. Theoretically we could start from Eqn.(4) as the condition of Theorem 1 and skip all steps above Eqn.(4). We set $P(X_{te}|X_{tr}) = P(X_k|X_{-k})$ as the condition because Theorem 1 provides insights for the proposed framework and y is not available when splitting data points into folds.
> 5. We agree with this issue. We addressed it to some degree in the Discussion (page 20), but we will revise this to focus on the p-value component more specifically. We will also add more discussion earlier, where the framework is introduced, to orient the reader toward the framework as a tool for the exploration in the paper but not as a final word on how to approach these scenarios.

---

> > ### Comment · Reviewer_T2Qn · 2023-06-27
> > **Response to clarifications**
> >
> > Apologies for taking a while to respond to your clarifications. I see that the other reviewers shared my concerns regarding readability of Theorem 1, confirming that it would benefit from re-formulation. Are you planning on updating a revised version of the pdf itself at some point now that all reviews are available? I think I follow the argumentation you provided here in point 4, but whether the proof is easy enough to understand for the readers can only be verified by checking how it looks like in the revised version.
> >
> > I agree that the paper has sufficient contributions despite the mathematical simplicity of IBCV, so the comment on that was merely a general remark and not a request for changing anything.
> >
> > There is no need for me to see the code, so simply confirming that it exists and will be made available is enough. In general, my impression is that you can upload code as a zip package as Supplementary material here as long as it has been anonymized in the same way as the paper itself, but from my perspective there is no need to do so now.

---

### Review · Reviewer_nnud · 2023-06-26

**Summary Of Contributions:**

The paper considers cross-validation in the context of geospatial data in which cross-validation procedures are challenged by spatial dependence and covariate shifts between training and test splits violating the common assumption of i.i.d. (independence and identically distributed) observations. The authors propose the IBCV and importance weighted buffered approach to CV and make extensive evaluations of various geospatial CV procedures on synthetically generated and real datasets. Apart from introducing the IBCV procedure the key contribution is extensive comparison of CV procedures systematically evaluating spatial dependency as opposed to spatial independence (SD vs SI) and covariate shift as opposed to no covariate shift (no CS vs. CS) in which the IBCV is favorable in the SI+CS scenario. Spatial dependence is assessed using spatial autocorrelation in training and test data whereas covariate shift is assessed using a non-parametric Cramér test. Importantly, the extensive analyses provided are used to make general recommendations in regards to best practices for geospatial cross-validation.

**Audience:**

Yes

**Broader Impact Concerns:**

There is in general no major ethical concerns of the executed research. The authors could emphasize on the importance of correct performance assessment not leading to erroneous conclusions, however, they do a good job in the introduction discussing and establishing this, while in the discussion discussing the broader importance of geo-spatial cross-validation.

**Claims And Evidence:**

Yes

**Requested Changes:**

It would be good to provide a clear recommendation how parameters should be tuned and perhaps also provide the reader with a recipe for recommendations according to the four scenarios SD/SI, no CS/CS in a box summarizing the key take-aways and guidelines of the paper.

The proof of theorem 1 is very condensed and difficult to follow. I suggest the steps in the derivation be elaborated and terms combined explicitly to derive P(T_te|T_Tr) and p(T_k|T_-k).

The importance weighting relies on some density estimation procedure for p_tr(x) and p_te(x). Presently the authors consider Relative unconstrained Least-Squares Importance Fitting (RuLSIF) to quantify these density ratios. It would be good to further discuss this choice as opposed to many alternatives providing density estimates as well as the impact of choice of procedure here chosen which I would expect can substantially impact results.

It is very unclear to me how the target test error is known for the real datasets considered as this has to be quantified by some cross-validation means as the generative process, noise structure etc. is unknown. Please clarify how the test error target value is quantified and known.


Minor comment: The paper in general reads well but please correct the following sentence:
response variables are the also same-> response variables are also the same



**Strengths And Weaknesses:**

Strengths:

*The IBCV procedure is interesting and provides a useful tool for geospatial cross-validation in particular for the SI+CS condition.

*The experimentation is extensive providing thorough synthetic and real data experimentation.

Weaknesses

* The proposed IBCV is rather straightforward combining IWCV and BFCV to do both. As such, the technical innovation of the paper is rather limited.

* The procedure relies on the tuning of hyperparameters and their careful tuning appear to be a challenge.

* The overall recommendations and results are somewhat non-surprising making the contribution limited although well executed and extensive.

---

> ### Author Response · Authors · 2023-07-06
> **Thanks for your comments and suggestions**
>
> We are almost ready to upload a new version of the paper, but in the meantime, here are a few thoughts.
>
> Responses to the noted weaknesses:
> 1. Yes, as discussed with another reviewer, we agree that IBCV is straightforward; that said, many widely-used cross-validation methods are simple. Despite its simplicity, it has not yet been discussed in the literature. We provide theoretical proof of the unbiasedness property and practical usefulness in a common scenario. We emphasize that IBCV is only one of our paper's contributions; we believe the overview of the CV settings and algorithms is also valuable.
> 2. We agree that the sensitivity to tuning parameters is unfortunate. This is true for the proposed IBCV as well as some existing methods. In future work, we hope to find a better solution, but for the moment, we aim to draw attention to this issue. We will keep this in mind in our revisions.
> 3. We observe that many practitioners take standard cross-validation for granted and ignore the characteristic of geospatial problems, resulting in poor model evaluation and model selection, which is one motivation of this research project. In addition, the recommendations are not quite predictable; some recent research papers draw controversial conclusions about the performance of various cross-validation methods (see the last second paragraph of Section 2).
>
> Responses to the requested changes:
>
> 4. We are working to clarify this in the revised version (coming soon). There will be some discussions about how to choose appropriate p-values for the Cramer test and hyperparameters of CV algorithms in section 8.
> 5. In the revised version, we are adding more details about how to derive from Eqn.(3) to Eqn.(4).
> 6. We chose RuLSIF for its advantage in theory and practice. uLSIF provides a closed-form solution computed by solving a system of linear equations while LSIF accumulates numerical errors, and thus uLSIF is more numerically stable than LSIF. In importance estimation and covariate shift adaptation tasks, uLSIF is empirically more accurate and faster than other four importance estimation algorithms - KDE, KMM, LogReg, and KLIEP.  Besides the shared advantages like analytic solution, numerical stability and robustness, RuLSIF improves on uLSIF, achieving an even better non-parametric convergence property. We are adding a brief comparison of these methods in section 2.
> 7. Great to know that this was unclear. In our experiments, a real dataset is divided into a training and test set. Each has known values of features and response variables. The ground truth test error is the classification error rate or RMSE computed by fitting the model to the training set and evaluating on the test set. We compare this ground truth against the estimate from cross-validation on the training set only. We will look for opportunities to clarify this. Please let us know if it still doesn’t make sense.
> 8. Thanks for pointing it out! We have fixed it in the revised version (coming soon).

---

> > ### Comment · Reviewer_nnud · 2023-07-12
> > **Thanks for clarifications and the revision**
> >
> > I thank the authors for clarifying my questions and their revisions of the manuscript taking these points into account.

---

### Decision · Action_Editors · 2023-09-08

**Recommendation:** Accept as is

**Comment:**

The reviewers support acceptance without much controversy. Having looked at the reviews and the updated version of the manuscript, I'm satisfied with the reply by the authors and support acceptance.

**Audience:**

This is indeed a very important topic not only for specific audiences interested in geospatial data and prediction models but also to trigger the need for thoughtful ways to validate performance in strongly correlated datasets.

**Claims And Evidence:**

Claims are well supported by an extensive set of simulated and real datasets, comparing against the proper baselines.